# BaSIS-Net: From Point Estimate to Predictive Distribution in Neural Networks - A Bayesian Sequential Importance Sampling Framework

**Giuseppina Carannante**                                                          *carannang1@rowan.edu*
*Department of Electrical and Computer Engineering*
*Rowan University*

**Nidhal C. Bouaynaya**                                                            *bouaynaya@rowan.edu*
*Department of Electrical and Computer Engineering*
*Rowan University*

**Ghulam Rasool**                                                                 *ghulam.rasool@moffitt.org*
*Machine Learning Department*
*Moffit Cancer Center*

**Lyudmila Mihaylova**                                                           *l.s.mihaylova@sheffield.ac.uk*
*Department of Automatic Control and Systems Engineering*
*The University of Sheffield*

**Reviewed on OpenReview:** *https://openreview.net/forum?id=V92PnXQ7UW*

## Abstract

Data-driven Deep Learning (DL) models have revolutionized autonomous systems, but ensuring their safety and reliability necessitates the assessment of predictive confidence or uncertainty. Bayesian DL provides a principled approach to quantify uncertainty via probability density functions defined over model parameters. However, the exact solution is intractable for most DL models, and the approximation methods, often based on heuristics, suffer from scalability issues and stringent distribution assumptions and may lack theoretical guarantees. This work develops a Sequential Importance Sampling framework that approximates the posterior probability density function through weighted samples (or particles), which can be used to find the mean, variance, or higher-order moments of the posterior distribution. We demonstrate that propagating particles, which capture information about the higher-order moments, through the layers of the DL model results in increased robustness to natural and malicious noise (adversarial attacks). The variance computed from these particles effectively quantifies the model's decision uncertainty, demonstrating well-calibrated and accurate predictive confidence.

## 1 Introduction

The promise to deploy Deep Learning (DL) models in real-world scenarios, while ensuring safe and reliable predictions, has led to an increasing focus on confidence quantification for DL models within the community (Tan et al., 2020; Kabir et al., 2018; Begoli et al., 2019; Dera et al., 2021; Carannante et al., 2020; Tabarisaadi et al., 2022; Abdar et al., 2021). The growing interest in uncertainty estimation is primarily driven by the brittleness of most point-estimate (or *deterministic*) models. These models produce a single real number, typically the output of the softmax function, as their prediction. Standard deterministic neural networks (NNs) perform remarkably well on data similar to what was used during training. However, they fail to generalize effectively to unfamiliar or perturbed data and may fail without any prior indication or warning

(Moosavi-Dezfooli et al., 2017; Nguyen et al., 2015; Biggio & Roli, 2018; Ahmed et al., 2022). Additionally, deterministic DL-based classifiers tend to assign high softmax scores to unrecognizable and out-of-distribution inputs or incorrect predictions without any visible signs of model failure (Moosavi-Dezfooli et al., 2017; Nguyen et al., 2015; Ahmed et al., 2022; Zhang & Li, 2019; Yuan et al., 2019; Carannante & Bouaynaya, 2023). The brittleness of DL models raises concerns about the reliability and trustworthiness of the predictions, particularly in situations where the model is to be deployed in mission-critical settings. When a DL model is integrated into a system that makes decisions affecting human life, either directly or indirectly, it is of paramount importance to determine whether the model is predicting reliably or failing randomly.

Point-estimate or deterministic models can erroneously interpret the softmax output value as the model's confidence or uncertainty information. Bayesian treatment of NNs allows for a formal definition of confidence or uncertainty, in the statistical sense (MacKay, 1992; Neal, 2012). In the Bayesian setting, a NN is treated as a probabilistic model, where the model parameters are viewed as random variables endowed with a probability density function (pdf) (MacKay, 1992; Neal, 2012). Using Bayes' rule, one can find the posterior pdf of the network's parameters given the training data. This posterior pdf encompasses all the information about the model and can be used with new data to find the predictive distribution. In particular, the second moment of the predictive pdf may provide a measure of the model's uncertainty attached to the prediction. However, due to the enormous parameter space and multiple layers of nonlinear functions in current DL models, exact inference in closed form is infeasible. Consequently, the community has resorted to various approximation techniques.

In the context of Bayesian deep learning (DL), where the model parameters are treated as random variables, several frameworks have been proposed to approximate the intractable inference problem (MacKay, 1992; Neal, 2012; Jordan et al., 1999; Graves, 2011; Blundell et al., 2015; Gal & Ghahramani, 2016; Kendall & Gal, 2017; Gal & Ghahramani, 2015; Cui et al., 2020; Lakshminarayanan et al., 2017; Roth & Pernkopf, 2016; Posch & Pilz, 2020; Carannante et al., 2020; 2021a; Dera et al., 2021).

Variational Inference (VI) is one of the most well-known approaches that defines an easy-to-estimate pdf, named *variational distribution*, and minimizes the Kullback–Leibler (KL) divergence between the true posterior distribution and the variational distribution (Jordan et al., 1999). Several Bayesian methods used VI, either by sampling a single set of parameters or by propagating the first two moments (mean and covariance) of the variational distribution during model training (Jordan et al., 1999; Graves, 2011; Blundell et al., 2015; Kendall & Gal, 2017; Gal & Ghahramani, 2015; Roth & Pernkopf, 2016; Carannante et al., 2020; Dera et al., 2021). Other frequentist approaches, such as Monte Carlo (MC) dropout and model ensemble, have been revisited and reinterpreted as approximate Bayesian inference for uncertainty estimation (Gal & Ghahramani, 2016; Kendall & Gal, 2017; Gal & Ghahramani, 2015; Lakshminarayanan et al., 2017; Hoffmann & Elster, 2021; Russell & Reale, 2021). VI approximates the inference problem into an optimization problem and is suited for large data sets. However, it is very hard to derive optimal guarantees or error bounds for the posterior pdf because the KL divergence between the variational distribution and the posterior cannot be calculated. Moreover, the ensemble approaches (Carannante et al., 2020), although able to capture higher-order statistics of the propagating distributions, are still constrained by the Gaussianity assumption.

In this paper, we present a novel approach to Bayesian DL that goes beyond the Gaussian approximation of the posterior distribution and model approximation. We formalize a statistical sequential importance sampling (SIS) framework along with a state-space formulation to estimate the predictive pdf of a DL model. Particle Filtering (PF), which relies on the SIS framework, was used in the statistical tracking community to estimate target distributions in non-linear and non-Gaussian dynamical systems (Crisan & Doucet, 2002). Unlike other methods, our proposed SIS framework can approximate any density, not only Gaussians, and has established performance guarantees (Crisan & Doucet, 2002). Additionally, the PF can be implemented in parallel for efficient computation. The power of the PF stems from its simple implementation and powerful convergence and optimality properties. Under weak assumptions, we can ensure (almost sure) convergence of the empirical distributions toward the true ones. Some bounds on the mean square errors and some large deviation results were elegantly and concisely presented in (Crisan & Doucet, 2002). Despite the power of the SIS method, to the best of our knowledge, no research has investigated the SIS framework within DL models and studied its uncertainty properties under random and adversarial conditions. A handful of research undertakings have been made in the classical machine learning literature to combine hidden state

estimation techniques with NNs(Singhal & Wu, 1988; Puskorius & Feldkamp, 1991; de Freitas et al., 2000; Halimeh et al., 2019; Li et al., 2010; Titensky et al., 2018; Carannante et al., 2021a). Most of these early attempts were focused on developing an alternative to the backpropagation algorithm for model training. These early efforts were hindered by the curse of dimensionality (the hidden state is used to represent the NN model parameters), and the applicability of these approaches remained limited to small NN models.

In this work, we propose a SIS framework and derive the mathematical formulation for estimating the predictive pdf of a DL model. We propose using a set of weighted particles to track and estimate the posterior pdf of the network's parameters as it *evolves* during model training. At the inference time, we estimate the complete predictive distribution of the unseen test data points, and the second central moment to determine the model's predictive uncertainty. The contributions of our work are:

- We introduce the SIS framework and PF method to propagate and learn the posterior pdf of the network's parameters. To address the issue of the curse of dimensionality, we employ multiple PFs along with a resourceful transition model that reduces the number of needed particles to achieve a reasonable estimate. At inference time, the learned particles are used as an approximation to the parameter's posterior pdf, and *no additional post-hoc sampling is required to compute the model uncertainty.*

- We conduct comprehensive robustness analysis using random noise and adversarial attacks. We compare the performance of our approach, named BaSIS-Net, with other state-of-the-art frameworks under various noisy conditions. We demonstrate the advantages of propagating higher-order moments, i.e., propagating and learning the full distribution with no constraint on the form of the posterior pdf.

- We propose and formalize an uncertainty metric to evaluate the consistency and meaningfulness of the obtained uncertainty estimates. Specifically, we propose a monotonic property, i.e., uncertainty monotonically increasing with the increasing levels of noise or severity of the attacks. We follow the general idea that a useful uncertainty estimate should convey lower confidence for the unknown, perturbed data.

The rest of the paper is organized as follows. Section 2 provides an in-depth literature review of Bayesian DL, uncertainty estimation approaches, and the use of PF and density estimation techniques for NN. In Section 3, we begin by providing a general overview of Bayesian learning and the SIS framework before proceeding to derive the proposed BaSIS framework for DL models, referred to as BaSIS-Net. Section 4, outlines the experimental setup, and Section 5 presents the results, followed by an elaborate discussion of the findings in Section 6. We conclude the paper with Section 7.

## 2 Related Work

### 2.1 Bayesian Neural Networks

Bayesian learning for NNs has seen a renewed interest in recent years due to the shifting interest in developing NNs capable of assessing their confidence along with their predictions. However, the exact Bayesian inference problem in NNs is intractable, so various approximation techniques have been developed to address this issue (MacKay, 1992; Neal, 2012; Jordan et al., 1999; Graves, 2011; Blundell et al., 2015; Gal & Ghahramani, 2016; Kendall & Gal, 2017; Gal & Ghahramani, 2015; Lakshminarayanan et al., 2017; Roth & Pernkopf, 2016; Carannante et al., 2020; 2021a; Dera et al., 2021). Among these, VI gained popularity and several variants have been proposed (Jordan et al., 1999; Graves, 2011; Blundell et al., 2015; Kendall & Gal, 2017; Gal & Ghahramani, 2015; Roth & Pernkopf, 2016; Posch & Pilz, 2020; Carannante et al., 2020; Dera et al., 2021). VI builds upon the idea of casting Bayesian inference as an optimization problem that can be solved with classical optimization techniques, such as gradient descent. A parameterized variational distribution is defined over the network's parameters, and the KL divergence between the defined distribution (variational distribution) and the unknown posterior distribution is minimized with respect to the parameters of the variational distribution. To compute the loss function, most approximating distributions are chosen from

a simple family of distributions, such as exponential. Early attempts at VI computed the Evidence Lower Bound (ELBO), which is equivalent to the KL divergence analytically (Jordan et al., 1999). However, this approach required conditionally conjugate distributions. To avoid this restriction, some researchers evaluated the ELBO with noisy estimates of the gradient, obtained by averaging Monte Carlo (MC) samples (Blundell et al., 2015). For a recent review of VI, we refer the reader to (Zhang et al., 2018).

In parallel with the development of VI, several researchers have focused on providing simple and scalable approximate Bayesian frameworks where little to no change is made to point-estimate networks (Gal & Ghahramani, 2016; Kendall & Gal, 2017; Gal & Ghahramani, 2015; Lakshminarayanan et al., 2017; Hoffmann & Elster, 2021). For instance, in MC Dropout, the dropout operation is re-formulated to approximate the variational distribution. by posing a Bernoulli prior distribution on the networks' weights or, in later work, as a mixture of multiple Gaussian pdfs (Gal & Ghahramani, 2016; 2015; Kendall & Gal, 2017). A similar approach is implemented with the model ensembling technique where a collection of models is obtained by training multiple NN models with distinct parameter initialization or training on separate mini-batches of the data (Lakshminarayanan et al., 2017). Some authors have tried to re-interpret model ensembling technique as Bayesian approximation in the same way as MC Dropout (Lakshminarayanan et al., 2017; Hoffmann & Elster, 2021).

Prior Networks (Malinin & Gales, 2018; 2019) offer an alternative to modeling uncertainty by allowing for data, distributional, and model uncertainty to be treated separately within a consistent, probabilistically interpretable framework. Prior Networks train a network to generate uncertainty estimates that differentiate between in-distribution and out-of-distribution (OOD) data. This differentiation is achieved by training the network on both in-distribution and OOD data. However, as highlighted in (Malinin & Gales, 2018), OOD samples are often not readily available, leading to the suggestion that one should either synthetically generate OOD data using a generative model or employ an alternate dataset (e.g., using FashionMNIST (Xiao et al., 2017) for MNIST (Deng, 2012)).

Most approaches built on VI and approximate Bayesian methods use a sampling technique to avoid propagating the entire pdf. Specifically, at each forward pass, only one sample is drawn from the approximating distribution. Instead, at the inference time, uncertainty is obtained by drawing multiple MC samples and performing multiple forward passes through the network, or by averaging the predictions of different networks.

## 2.2 Density Propagation Methods for Neural Networks (NNs)

Estimating the posterior distribution of the model parameters can also be cast from a dynamical system perspective. The problem of estimating the hidden state of a stochastic dynamical system based on noisy measurements is a fundamental problem in hidden-state tracking and estimation. The state-space formulation can describe several, and NN training can be modeled as a non-linear discrete-time system. Several authors have used tracking methods, e.g., Kalman filters (KFs) or PFs, to explore alternative training approaches for NNs (Singhal & Wu, 1988; Puskorius & Feldkamp, 1991; de Freitas et al., 2000; Halimeh et al., 2019; Li et al., 2010; Titensky et al., 2018; Carannante et al., 2021a; Russell & Reale, 2021). However, these approaches have not gained popularity due to their limited applicability to small and simple (fully-connected) NNs, and their evaluation has primarily focused on small, clean datasets with little study on related uncertainty quantification. Only two studies, (Titensky et al., 2018; Carannante et al., 2021a), have applied these techniques to quantify uncertainty in small fully-connected (FC) networks trained on the MNIST dataset (Deng, 2012), while (Russell & Reale, 2021) focuses on multivariate uncertainty in the context of Bayes filtering systems.

Recent works adopted the VI framework to track the first two posterior moments, i.e., mean and covariance, across the non-linear layers of a network, levereging key techniques from extended, ensemble and unscented KF (Dera et al., 2021; Carannante et al., 2020) . Unlike other VI approaches, the moments propagation approach allows for the learning of uncertainty information during model training. We refer to these tracking approaches that attempt to track the moments of the posterior pdf across the model layers and rely on the Vi framework as *Variational Density Propagation (VDP)*. It is important to note that VDP is limited to Tensor

Normal distributions due to the use of VI and the KL divergence, resulting in a closed-form expression for the loss function (Dera et al., 2021).

This paper goes beyond the VDP approach, i.e., beyond the first two moments and the VI approximation, by utilizing the Sequential Importance Sampling (SIS) and Particle Filtering (PF) techniques, which provide asymptotic convergence guarantees, to estimate the posterior pdf of the parameters and measure the prediction uncertainty of the model. Unlike previous work that leveraged the state-space model in NNs, our focus is not on finding an alternative to backpropagation but on estimating the posterior and predictive pdfs of the network's parameters for the test dataset. We refine these pdfs in a sequential manner as training progresses using weighted particles. Unlike the Variational Inference (VI) framework, which typically constrains the posterior and predictive pdfs to a predefined family of distributions, our approach employs a particle-based representation that offers a more flexible empirical approximation of these distributions. However, it is important to note that this flexibility is contextual and operates within the confines of the chosen transition and observation models. Additionally, we utilize the learned particles to derive and analyze uncertainty information.

## 3 Mathematical Formulation

### 3.1 Bayesian Framework

In the Bayesian setting, the model parameters $\mathcal{W}$ are defined as random variables with a prior distribution $p(\mathcal{W})$. In the Bayesian framework, all information about the (unknown) parameters is contained in the posterior distribution given the training data, $\mathcal{D} = \{(\mathbf{X}, \mathbf{y})_j\}_{j=1}^{M}$, i.e., $p(\mathcal{W}|\mathcal{D})$. In particular, we can derive the predictive distribution of any unseen input $\tilde{\mathbf{X}}$, with output $\tilde{\mathbf{y}}$, as follows:

$$p(\tilde{\mathbf{y}}|\tilde{\mathbf{X}}, \mathcal{D}) = \int p(\tilde{\mathbf{y}}|\tilde{\mathbf{X}}, \mathcal{W}) \; p(\mathcal{W}|\mathcal{D}) \; d\mathcal{W}. \tag{1}$$

In equation (1), the predictive distribution's mean corresponds to a point estimate prediction, whereas the second-order moment can be utilized to assess the model's confidence. Note that the predictive distribution depends on the evaluation of the posterior pdf of the model parameters $p(\mathcal{W}|\mathcal{D})$. Using Bayes rule, the posterior is derived as:

$$p(\mathcal{W}|\mathcal{D}) = \frac{p(\mathcal{D}|\mathcal{W})p(\mathcal{W})}{\int p(\mathcal{D}|\mathcal{W})p(\mathcal{W})d\mathcal{W}}. \tag{2}$$

Where $p(\mathcal{D}|\mathcal{W})$ is the likelihood of the data given the parameters. However, as we will show in the sequel, a closed form solution of Eq. (2) is only possible in few simple cases.

### 3.2 State-Space Formulation

We consider the following state-space model for the model parameters:

$$\begin{aligned} \mathcal{W}_k &= g(\mathcal{W}_{k-1}) + \eta_k, \\ \mathbf{y}_k &= f(\mathcal{W}_k, \mathbf{X}_k) + v_k, \end{aligned} \tag{3}$$

where $\mathbf{X}_k$ is the $k^{\text{th}}$ input training sample with label $\mathbf{y}_k$, $\eta_k$ and $v_k$ are zero-mean process and measurement noise, respectively, with known pdfs. The non-linear maps $G$ and $F$ represent the state-transition and measurement models, respectively.

The Bayesian solution requires finding the posterior pdf of the state given the noisy measurements, $p(\mathcal{W}_k|\mathbf{y}_k)$, at every time step $k$. Using Bayes rule, we can derive the two-step Bayesian recursion formulas to recursively estimate the posterior as follows:

$$\begin{aligned} p(\mathcal{W}_k|\mathbf{y}_{k-1}) &= \int p(\mathcal{W}_k|\mathcal{W}_{k-1}) \; p(\mathcal{W}_{k-1}|\mathbf{y}_{k-1}) \; d\mathcal{W}_{k-1}, \\ p(\mathcal{W}_k|\mathbf{y}_k) &= \frac{p(\mathbf{y}_k|\mathcal{W}_k) \; p(\mathcal{W}_k|\mathbf{y}_{k-1})}{\int p(\mathbf{y}_k|\mathcal{W}_k) \; p(\mathcal{W}_k|\mathbf{y}_{k-1})}. \end{aligned} \tag{4}$$

The recursive relations given in Eq. (4) are only a conceptual solution due to the fact that these integrals are generally intractable. For the linear Gaussian model, the posterior is also a Gaussian distribution whose mean and covariance can be computed using the Kalman filter (KF) (Li et al., 2015a). In other cases, several approximations can be used, such as the extended Kalman filter (EKF) (Kalman, 1960), the unscented Kalman filter (UKF) (Julier & Uhlmann, 1997), and the ensemble Kalman filter (EnKF) (Evensen, 1994). However, the EKF, UKF, and EnKF all make different assumptions about the nature of the nonlinear models and all assume that the process and measurement noise are Gaussian.

### 3.3 Sequential Importance Sampling (SIS) and the Particle Filter (PF)

In non-linear and non-Gaussian state-space models, PF represents a powerful class of sequential MC algorithms that numerically solve the optimal estimation problem without making any assumptions about the pdfs or the linearity of the system model (Gordon et al., 1993; Doucet et al., 2001). PF approximates the unknown posterior pdf by a set of weighted samples, called particles. PF builds upon the *importance sampling* technique, which is used to approximate the expectation of a function with respect to a pdf that is difficult or impossible to sample from directly. The basic idea behind importance sampling is to transform the target distribution into an easier-to-sample distribution, called the *importance distribution*, and use this importance distribution to generate samples that are used to approximate the desired expectation. Theoretically, the only condition on the importance distribution is that its support includes the support of the posterior distribution.

Put in context, given the unknown posterior distribution, $p(\mathcal{W}_k|\mathbf{y}_k)$, from which it is difficult to sample from, we introduce an "easy to sample from" importance distribution $q(\mathcal{W}_k|\mathcal{W}_{k-1}, \mathbf{y}_k)$. We draw samples from the importance distribution, $\mathcal{W}_k^{(i)} \sim q(\mathcal{W}_k|\mathcal{W}_{k-1}, \mathbf{y}_k)$, associate an importance weight $u^{(i)}$ to make up the difference between the importance distribution and the posterior and find a discrete approximation of the posterior pdf as follows:

$$\hat{p}(\mathcal{W}_k|\mathbf{y}_k) = \sum_{i=1}^{N} u^{(i)} \delta(\mathcal{W}_k - \mathcal{W}_k^{(i)}), \tag{5}$$

where $\delta$ represents the Dirac function and $N$ is the number of particles. $\mathcal{W}_k^{(i)}$ denotes the $i$-th particle at time step $k$. The importance weights "calibrate" the particles, i.e., they carry information about how likely the particles - drawn from the importance distribution - are actually coming from the true target distribution. The posterior distribution is approximated by a set of Dirac-delta masses (particles) that evolve randomly in time according to the state-space model in (3). Additionally, it can be seen from Eq. (5) that the PF enables the computation of the expectation $\mathbb{E}(\cdot)$ of any test function $\phi$ as:

$$\mathbb{E}(\phi(\mathcal{W})) \approx \sum_{i=1}^{N} u^{(i)} \phi(\mathcal{W}_k^{(i)}). \tag{6}$$

Throughout the rest of this manuscript, we refer to the weights $u^{(i)}$ associated with particles as *importance ratios*, or simply *ratios*, to avoid any confusion with the *weights* of the DL model.

In the PF framework, the posterior is approximated sequentially by alternating an importance step, i.e., drawing particles from the importance distribution, $\mathcal{W}_k^{(i)} \sim q(\mathcal{W}_k|\mathcal{W}_{k-1}, \mathbf{y}_k)$, followed by an update step where the particles are adjusted by taking into account their likelihood, i.e., computing the importance ratios. Specifically, the ratios are computed as:

$$u_k^{(i)} \propto u_{k-1}^{(i)} \frac{p(\mathbf{y}_k|\mathcal{W}_k^{(i)}) p(\mathcal{W}_k^{(i)}|\mathcal{W}_{k-1}^{(i)})}{q(\mathcal{W}_k^{(i)}|\mathcal{W}_{k-1}, \mathbf{y}_k)}, \tag{7}$$

where $\propto$ denotes a proportionality relation. The importance ratios are then normalized to sum up to unity. The choice of the importance distribution can impact the rate of convergence of the filter (Doucet et al., 2000). The optimal importance pdf, which minimizes the variance of the importance ratios, is usually not accessible. A commonly used alternative, for its simplicity, is the state transition pdf $p(\mathcal{W}_k^{(i)}|\mathcal{W}_{k-1}^{(i)})$, which

reduces the computation of the ratios to:

$$u_k^{(i)} \propto u_{k-1}^{(i)} \; p(\mathbf{y}_k | \mathcal{W}_k^{(i)}). \tag{8}$$

Equation (8) indicates that if the importance distribution is selected as the prior transition, the importance ratios become directly proportional to the likelihood.

The power of the PF stems not only from its simple implementation but also from its powerful convergence and optimality properties. Under weak assumptions, we can ensure (almost sure) convergence of the empirical distribution toward the true one (as the number of particles increases). Some bounds on the mean square errors and some large deviation results were elegantly and concisely presented in (Crisan & Doucet, 2002).

In practical implementations, the number of particles $N$ is finite, and the PF provides estimates whose variance increases with time. However, resampling largely solves this problem (Li et al., 2015b). Resampling is a very intuitive idea, which has major practical and theoretical benefits (Doucet et al., 2001). An important advantage of resampling is that it allows us to remove particles with low weights and thus save on the computational cost of propagating them. A systematic resampling technique consists in replicating particles with larger weights while eliminating those with smaller weights. Many resampling schemes have been proposed in the literature. However, the systematic resampling is the most widely-used algorithm as it is extremely easy to implement and outperforms other resampling techniques in most scenarios (Doucet et al., 2001).

As with any scheme for numerical integration, be it deterministic or stochastic, the characterization of the distribution with a fixed number of samples deteriorates in sufficiently high-dimensional spaces and with sufficiently complex distributions (Bengtsson et al., 2008). The number of samples required increases exponentially with the dimensionality of the problem. Efficient high-dimensional importance sampling techniques and conditions were investigated in the literature (Bengtsson et al., 2008; Richard & Zhang, 2007; Beck, 2003). As previously mentioned, the choice of importance distribution can significantly impact the convergence of the PF. Some authors have proposed including likelihood terms in the importance distribution to reduce the number of required particles (Ades & Leeuwen, 2013). Alternatively, others have proposed performing multiple PFs by partitioning the high-dimensional state into sub-states (Djuric et al., 2007; Closas & Bugallo, 2012). For a complete and detailed treatment of PF algorithms, importance density choice, mathematical convergence derivation and different resampling schemes proposed in the literature, we refer the reader to the surveys and tutorials in (Arulampalam et al., 2002; Doucet & Johansen, 2009; Crisan & Doucet, 2002; Li et al., 2015b).

### 3.4 BaSIS-Net

The DL model is modeled as a discrete-time dynamical system with the hidden state represented by the network's parameters $\mathcal{W}$. Consequently, the terms "state" and "parameters" are used interchangeably. We let $N$ denote the number of particles, $k = 1, \cdots, K$ represent the time-step, where $K$ is typically determined by the number of epochs multiplied by the number of batches. $l$ refers to the layer, with $L$ being the total number of layers in the network. The posterior distribution of the weights is approximated by a set of Dirac-delta masses (samples) that evolve across the layers according to the dynamics of the training data. The sequential derivation of the posterior requires us to determine the transition distribution of the weights $p(\mathcal{W}_k | \mathcal{W}_{k-1})$ and the likelihood $p(\mathbf{y}_k | \mathcal{W}_k)$.

First, we break the high-dimensional system into smaller-dimensional systems (subsystems) by partitioning the state at time $k$ into $L$ components, each corresponding to a layer $l$ as follows: $\mathcal{W}_k = \{\mathcal{W}_{[1]k}, \cdots, \mathcal{W}_{[L]k}\}$. The exponent will be reserved to denote a sample. We subsequently adopt the concept of multiple or parallel particle filters as in Djuric et al. (2007).

We introduce a well-crafted transition model where the evolution of the state is guided by the gradient descent update rule. Specifically, the map $g$ in (3) is defined as:

$$\mathcal{W}_k = \mathcal{W}_{k-1} - \alpha \nabla_{\mathcal{W}_{k-1}} \mathcal{L}(\mathcal{W}_{k-1}), \tag{9}$$

where $\nabla_{\mathcal{W}} \mathcal{L}$ is the gradient of the network loss function with respect to the parameters, and $\alpha$ is the learning rate. As described in the Section 3.3, we use the transition density as our importance density.

An intuitive example of a likelihood function is given by the "distance" between the network prediction, given the current sample weights, and the target output. Given the subsystem or layer-wise state calculations, the output of the network is calculated using the current sample particles for the considered layer while all other layers are frozen at their most recent estimates. Formally, we have

$$p(\mathbf{y}_{[l]k}|\mathcal{W}_k^{(i)}) \propto \exp\left(-\mathcal{L}(\mathbf{y}_{[l]k}^{(i)})\right), \tag{10}$$

where

$$
\begin{aligned}
\mathbf{y}_{[l]k}^{(i)} &= f([W_k]^{(i)}, \mathbf{X}), \text{ and} \\
[W_k]^{(i)} &= \{\mathcal{W}_{[1]k}, \cdots, \mathcal{W}_{[l-1]k}, \mathcal{W}_{[l]k}^{(i)}, \mathcal{W}_{[l+1]k-1}, \cdots, \mathcal{W}_{[L]k-1}\}.
\end{aligned}
\tag{11}
$$

Algorithm 1 outlines the specific steps for implementing the learning of the posterior pdf and the estimation of the predictive pdf. To reduce computational complexity, we compute the gradient of the loss with respect to the previous time step weighted mean estimate rather than for each individual particle. Furthermore, a simple resampling technique is utilized in which particles survive in proportion to their likelihood, and offspring particles are assigned an importance ratio of $1/N$. As a result, it is not necessary to store importance ratios.

---

**Algorithm 1** BaSIS

---

1: Let $\mathcal{W}_0 = \{\mathcal{W}_{[1]0}, \cdots, \mathcal{W}_{[L]0}\}$ be the partitioned state at time $k = 0$.
2: For $k = 0$, draw particles from the prior pdf: $\mathcal{W}_{[l]0}^{(i)} \sim p(\mathcal{W}_{[l]0})$, e.g., a Gaussian or Uniform, and assign uniform importance ratios: $u_{[l]0}^{(i)} = 1/N$ for all $l = 1, \cdots, L$ and $i = 1, \cdots, N$.
3: **for** $k = 1, 2, \ldots, K$ **do**
4:     **for** $l = 1, 2, \ldots, L$ **do**
5:         Sample $\mathcal{W}_{[l]k}^{(i)} = \mathcal{W}_{[l]k-1}^{(i)} - \alpha \nabla_{\mathcal{W}_{[l]k-1}} \mathcal{L}(\mathcal{W}_{k-1}) + \eta^{(i)}, \ \eta^{(i)} \sim \mathcal{N}(0, \sigma_\eta^2 \mathbf{I}), \ i = 1, 2, \ldots, N$
6:         Evaluate $\mathbf{y}_{[l]k}^{(i)} = f([W_k]^{(i)}, \mathbf{X})$, where $[W_k]^{(i)} = \{\mathcal{W}_{[1]k}, \ldots, \mathcal{W}_{[l-1]k}, \mathcal{W}_{[l]k}^{(i)}, \mathcal{W}_{[l+1]k-1}, \ldots, \mathcal{W}_{[L]k-1}\}$, $i = 1, 2, \ldots, N$
7:         Compute $\tilde{u}_{[l]k}^{(i)} \propto u_{[l]k-1}^{(i)} e^{-\mathcal{L}(\mathbf{y}_{[l]k}^{(i)})}$
8:         Normalize importance ratios $u_{[l]k}^{(i)} = \frac{\tilde{u}_{[l]k}^{(i)}}{\sum_j \tilde{u}_{[l]k}^{(j)}}, \ i = 1, 2, \ldots, N$
9:         Resample and associate a weight of $1/N$ with each offspring sample
10:        Calculate $\mathcal{W}_{[l]k} \leftarrow \frac{1}{N} \sum_i \mathcal{W}_{[l]k}^{(i)}$
11:     **end for**
12:     Update $\mathcal{W}_k \leftarrow \{\mathcal{W}_{[1]k}, \cdots, \mathcal{W}_{[L]k}\}$
13: **end for**

---

### 3.4.1 Basis Inference Mechanism

After learning the optimal particles, we can directly compute the full predictive distribution for any given input without needing further sampling. From this distribution, its associated moments, such as the mean for point-estimate classification and the variance for predictive uncertainty, can be calculated without additional post-hoc sampling.

Without loss of generality, we demonstrate the propagation for a two-layer network. Let $\mathbf{y} = f_2(W_{[2]}, (f_1(W_{[1]}, \mathbf{X}))$ be the output of the network, where $\mathbf{X}$ is the input, $W_{[1]}, W_{[2]}$ are the parameters for the two layers and $f_1, f_2$ denote the network output at layers 1 and 2, respectively. Let $\{\mathcal{W}_{[1]K}^{*(i)}, \mathcal{W}_{[2]K}^{*(i)}\}_{i=1}^N$

be the $N$ learned particles, i.e., obtained at the final training step $K$. To simplify the notation, we will drop the step notation $K$.

At inference/test time and given an input $\mathbf{X}$, we first propagate the particles through the first layer $f_1$:

$$A^{(i)} = f_1(\mathcal{W}_{[1]}^{*(i)}, \mathbf{X}), \quad i = 1, \dots, N, \tag{12}$$

where $\{A^{(i)}\}_{i=1}^N$ is the obtained probability distribution after the first layer. Then, we compute the sample mean, $\bar{A}$, of this distribution, using Eq.(6), and feed it to the next layer $f_2$:

$$\mathbf{y}^{(i)} = f_2(\mathcal{W}_{[2]}^{*(i)}, \bar{A}), \quad i = 1, \dots, N, \tag{13}$$

where $\{\mathbf{y}^{(i)}\}_{i=1}^N$ is an empirical approximation of the predictive distribution. The first-order moment corresponds to the point-estimate prediction, and the second-order moment quantifies the associated predictive uncertainty.

### 3.4.2  BaSIS-Net Hyperparameters

PF methods' performance is linked to the number of particles $N$ used. Theoretically, PF approaches converge to the optimal filter as $N \to \infty$ (Crisan & Doucet, 2002). At time $t = 0$, Basis-Net is initialized by drawing particles from a prior pdf: $\mathcal{W}_{[l]0}^{(i)} \sim p(\mathcal{W}_{[l]0})$, e.g., a Gaussian or Uniform. The parameters of this distribution, e.g., $\sigma$ for Gaussian distributions, should be treated as other hyperparameters of NN model. Hence, they should be optimized.

In the general context of state space models, the noise values in the transition model govern the search space for particles at the subsequent time step. As has been well-established since the advent of the Kalman filter, the selection of $\sigma_\eta$ impacts the algorithm's convergence behavior (Gordon et al., 1993; Park et al., 2013). When selecting the sigma parameter for the noise in the transition model of the PF, it is crucial to consider the trade-off between exploration and exploitation (or convergence). Given that in practice, we have a finite number of particles, using a large sigma encourages exploration of the state space but may hinder convergence. Conversely, a small sigma restricts the search space, reducing particle diversity and potentially compromising the effectiveness of the filtering/tracking process.

When starting our experiments, for each dataset, we performed a grid-search over a set of values for $\sigma_\eta$, starting from large to small, to identify a value that guarantees convergence with a good test accuracy. In the Appendix A, we also report the accuracy of BaSIS-Net versus $\sigma_\eta$ on the MNIST and CIFAR-10 dataset as a proof of concept. Additionally, we investigated the robustness and uncertainty calibration of the model with varying $\sigma_\eta$. Figure 11 shows test accuracy and predictive variance under Gaussian noise applied to MNIST test data, with several BaSIS-Net models using different $\sigma_\eta$ values.

## 4  Experimental Setup

### 4.1  Datasets and Model Architectures

The performance of the proposed BaSIS-Net framework is tested and evaluated on four distinct datasets. We start our experiments with two standard image classification benchmark datasets, MNIST and CIFAR-10 (Deng, 2012; Krizhevsky & Hinton, 2009). Subsequently, we test the proposed framework on two medical image datasets. The first medical task involves image classification using the Malaria dataset(Rajaraman et al., 2018), which is publicly accessible via the Tensorflow Datasets Catalog. The dataset consists of $27,558$ thin blood smear slide images of both infected (parasitized) and uninfected cells. The images in this dataset are highly diverse and heterogeneous, captured using various microscopic scanners. As a pre-processing step, images are zero-padded to achieve a uniform input dimension of $200 \times 200$ when required.

In the second medical task, the focus is on image segmentation using the Hippocampus dataset from the Medical Segmentation Decathlon (Simpson et al., 2019; Antonelli & et al., 2022). The task consists of segmenting two adjacent structures of the hippocampus, i.e., anterior and posterior regions. The Hippocampus dataset

includes 394 single-modality MRI scans of varying sizes. As a pre-processing step, scans are standardized to a uniform input size of $64 \times 64$ pixels through zero-padding, if necessary. Additionally, to minimize the bias inherent in MRI scans, all pixel values are normalized between 0 and 1.

**MNIST:** We use a small neural network comprising a single convolutional layer followed by Rectified Linear Units (ReLU) activation, max-pooling, a single fully-connected (FC) layer, and a final softmax operator. The convolutional layer consists of 32 kernels with a size of $5 \times 5$, while the FC layer comprises 10 neurons.

**CIFAR-10:** We use an architecture comprising 11 layers, consisting of 10 convolutional layers and one FC layer with 10 units. The convolutional layers contain 32, 32, 32, 32, 64, 64, 64, 128, 128, 128 kernels, respectively. The first block of operations is composed of convolutional kernels of size $5 \times 5$ followed by batch-normalization, Exponential Linear Unit (eLU) activation function, and max-pooling. The subsequent 4 blocks of operations consist of two convolution operations, each followed by eLU activation and batch-normalization, one max-pooling, and one dropout layer with a rate of 0.2. Convolutions are performed using kernels of size $3 \times 3$. The final block of operations comprises a single $1 \times 1$ convolution operation, eLU activation, batch-normalization, a final FC layer, and the softmax function.

**Malaria Classification Dataset:** We use a NN architecture consisting of seven convolutional layers and one FC layer. The first block of operations comprises a convolutional operation, followed by eLU activation, batch-normalization, and max-pooling. The subsequent three blocks are composed of a sequence of operations that include convolution, eLU activation, batch-normalization, another convolution operation, eLU activation, batch-normalization, max-pooling, and a dropout operation with a rate of 0.2. The number of kernels in each convolutional layer is set to 16, 16, 16, 16, 32, 32, and 32. The kernels in all layers have a size of $3 \times 3$. Finally, the architecture concludes with an FC layer with a softmax activation function.

**Hippocampus:** We use U-Net, an encoder-decoder architecture designed for segmentation tasks Ronneberger et al. (2015). The encoder path consists of three encoder block, each consisting of two convolutional layers followed by a ReLU activation and a max-pooling operation. The bottom encoder block does not perform max-pooling. The encoder has six layers, with the number of kernels set to 32, 32, 64, 64, 128, and 128 from top to bottom. All kernels have a size of $3 \times 3$. The decoder path is composed of two decoder blocks. Each decoder block performs an upsampling operation followed by a learnable $2 \times 2$ convolution, two consecutive convolutions $3 \times 3$ kernels, each followed by a ReLU activation and padding operations. The decoder has six layers, with the number of kernels set to 64, 64, 64, 128, 128, and 128 from bottom to top. The output is obtained by applying a $1 \times 1$ convolution and softmax to the decoded feature maps to obtain a segmentation mask of the same shape as the input. Similar to the original U-Net implementation, we concatenate the data from each encoder block with the corresponding decoder block.

## 4.2 DL Models

**Deterministic (Det)**: Det is a neural network model trained deterministically. This model generates deterministic predictions, meaning it produces a single prediction for each input without providing any indication of uncertainty.

**MC Dropout (MC Drop):** MC Drop is a neural network model that applies dropout not only during training but also during inference. It involves obtaining multiple predictions by performing several forward passes through the network during inference. These predictions are interpreted as samples from the approximate posterior distribution, and their average is used as the estimate. In all datasets and experiments, we use 20 samples. For classification tasks, the dropout probability is set to $p = 0.2$. For the segmentation dataset, we follow the specification in (Kendall et al., 2015) and apply dropout only at the bottom layers of the encoder-decoder network with a dropout probability of $p = 0.5$.

**Bayes Convolutional NN (BCNN):** BCNN is the extension of Bayes-By-Backprop (BBB) to Convolutional Neural Networks (CNNs) (Blundell et al., 2015; Shridhar et al., 2018). The model is trained within the Variational Inference (VI) framework using a Gaussian variational distribution. A single set of parameters is sampled at each forward pass during training. At inference time, the final prediction is obtained as the average of multiple forward passes through the NN, treated as samples drawn from the approximated posterior pdf. Similar to MC Drop, we consider 20 samples for all datasets and experiments.

**Ensemble Model:** For each dataset, we train a set of five standard NNs with the same architecture. The diversity among the ensemble members is achieved by considering different initialization schemes for the learnable parameters. At inference time, we feed the same input to all five trained NNs and obtain the ensemble model output as the average across all NNs (Lakshminarayanan et al., 2017).

**Variational Density Propagation (VDP) Models:** We consider three VDP frameworks, namely, (1) extended VI (exVI), (2) ensemble VI (enVI), and (3) unscented VI (unVI) (Dera et al., 2021; Carannante et al., 2020). Model training is performed by propagating the first two moments of the variational pdf, i.e., the mean and covariance. Depending on the approach employed (exVI, enVI or unVI), the propagation through activation functions is performed using either first-order Taylor series, ensembles, or unscented transformation. For all three approaches, we define a Tensor Normal Distribution over the parameters of the model. To reduce computational burden, for the segmentation task, we only propagate diagonal values of the covariance matrices (Carannante et al., 2021b). Due to the nature of VDP, no sampling is required at inference time.

**Bayesian Sequential Importance Sampling (BaSIS-Net):** We use BaSIS-Net to refer to DL models that employ the BaSIS framework. The number of particles utilized in a BaSIS model is indicated at the end, such as BaSIS 100, which is a BaSIS-Net with 100 particles. In all simulations, we set the prior pdf to a Gaussian distribution with $\sigma_{init} = 0.001$ for all detests. To generate particles, we set $\sigma_\eta = 0.0001$ for MNIST and Malaria datasets, $\sigma_\eta = 0.00001$ for CIFAR-10, and $\sigma_\eta = 0.00005$ for the Hippocampus dataset. Different numbers of particles are employed for each dataset. For MNIST, we use $N = 100$, 500, and 1000, for CIFAR-10, we use $N = 100$, and 500, and for the Hippocampus and Malaria datasets, we use $N = 100$. We adopt the cross-entropy loss function for $\mathcal{L}$ in Eq. (10) for all experiments. In all BaSIS-Net models, the particles are learned during training, and no *post-hoc* sampling is needed at inference time.

### 4.3 Performance Metrics and Robustness Analysis

We conducted a comprehensive robustness analysis for all models by evaluating their performance on datasets perturbed with various types of additive natural and adversarial noises. We computed the average accuracy for classification tasks and the Dice Similarity Coefficient (DSC) values for the segmentation task, with DSC values reported separately for both anterior and posterior structures for the Hippocampus dataset (Menze et al., 2014). The level of additive noise or perturbation was measured using Signal-to-Noise Ratio (SNR) in decibels (dB), with high SNR values indicating low noise/perturbation levels and vice versa. In the Results Section 6, we reported the average accuracy or DSC values for each model for the noise-free cases, i.e., model performance on the standard test datasets, and for different levels of additive noise and adversarial attacks. The best-performing model(s) were indicated in bold notation. In all tables and plots, we constructed the 95% Confidence Intervals for our results.

To generate adversarial noise, we used two well-known techniques, Fast Gradient Sign Method (FSGM) and Projected Gradient Descent (PGD) (Liu et al., 2016; Madry et al., 2017). We reported the attack type, targeted or untargeted, and the fooling class for each targeted experiment. For PGD attacks, we also reported the step-size and maximum number of iterations (Max-iter). Specifically, for the MNIST dataset, we generated targeted and untargeted FGSM attacks and used class label 3 as the fooling class for targeted attacks. For the CIFAR-10 dataset, we generated both targeted and untargeted FGSM and PGD attacks. For targeted FGSM, we set the fooling class to label 3, and for PGD, we considered two different fooling classes, i.e., label 8 and label 3, and changed the number of iterations to 10 and 40. For untargeted attacks, we changed the number of iterations only. For the Malaria dataset, we generated untargeted FGSM and PGD attacks, with the number of iterations set to 10 and the step size to 1 for PGD attacks. For the Hippocampus dataset, we only considered PGD attacks, with class label 2 as the fooling class and the number of iterations set to 5 for targeted attacks.

The study aims to showcase various attacks (FGSM and PGD targeted and untargeted) across diverse datasets (MNIST, CIFAR-10, Malaria, and Hippocampus), recognizing that attack effectiveness depends on the power of attack as well as inherent dataset characteristics. Experiments varied the attack intensity (measured by SNR) from high to low to comprehensively assess model resilience. We systematically explored

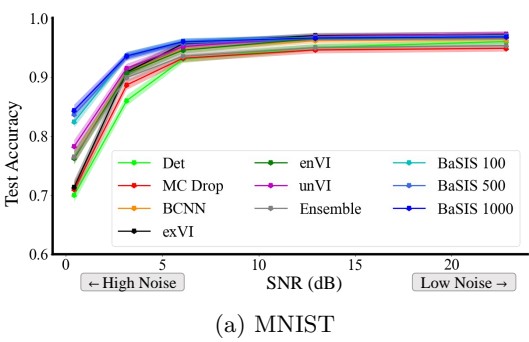
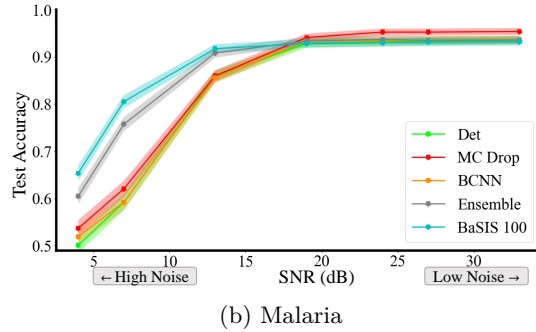

|     |     |
| :-: | :-: |
| (a) MNIST | (b) Malaria |

Figure 1: Performance of different models under various levels of Gaussian noise applied to (a) MNIST and (b) Malaria test dataset. The *y*-axis displays the accuracy at test time and *x*-axis represents the level of additive Gaussian noise measured as Signal to Noise Ratio (SNR) in units of dB. We note that at high noise levels, BaSIS-Net performs better than other models.

Table 1: Performance comparison using average test accuracy of different models under additive Gaussian Noise for CIFAR-10.

| Model | Det | MC Drop | BCNN | exVI | Ensemble | BaSIS 100 | BaSIS 500 |
| :---: | :---: | :---: | :---: | :---: | :---: | :---: | :---: |
| Noise Free | .83 ± .007 | .84± .007 | .83 ± .007 | **.86**± .007 | .85± .007 | .84± .007 | .84± .007 |
| SNR | | | | Gaussian Noise | | | |
| 28 low noise | .82 ± .008 | .82 ± .008 | .81 ± .008 | **.85** ± .007 | .83 ± .007 | .83 ± .007 | .83 ± .007 |
| 14 medium noise | .60 ± .010 | .70 ± .009 | .61 ± .010 | .61 ± .010 | .65 ± .009 | **.72** ± .009 | **.72** ± .009 |
| 4 high noise | .25 ± .010 | .27 ± .008 | .25 ± .008 | .25 ± .007 | .25 ± .007 | .30 ± .007 | **.31** ± .007 |

a variety of attack scenarios by employing a mix of targeted and untargeted attacks across different datasets. This approach allowed us to showcase a broad range of adversarial situations.

### 4.4 Training Hyperparameters and Settings

We employ different loss functions for various DL models. Specifically, we use cross-entropy loss function with $L2$ regularization for Det, MC Drop, Ensemble, and BaSIS, while employing ELBO loss function for VI variants. In the case of MNIST, we train all models for 10 epochs with a learning rate of 0.01. For CIFAR-10, we train models for 300 epochs with an initial learning rate of 0.01, apply polynomial decay, and early stopping to avoid overfitting. We use a batch size of 50 for both MNIST and CIFAR-10. For the Malaria dataset, we train all models for 20 epochs with polynomial decay and an initial learning rate of 0.01, with a batch size of 20. Similarly, for the Hippocampus dataset, we train all models for 150 epochs with Adam optimizer, a learning rate of 0.001, and a batch size of 10. We perform all simulations on an NVIDIA RTX A6000 GPU. For Malaria and Hippocampus datasets, we cannot train BaSIS using a higher number of particles due to GPU memory constraints.

Table 2: Performance comparison using average DSC of different models under additive Gaussian Noise for Hippocampus Dataset.

| | Anterior Structure of Hippocampus | | | | | Posterior Structure of Hippocampus | | | | |
| :---: | :---: | :---: | :---: | :---: | :---: | :---: | :---: | :---: | :---: | :---: |
| Model | Det | MC Drop | exVI | Ensemble | BaSIS-100 | Det | MC Drop | exVI | Ensemble | BaSIS-100 |
| Noise Free | .74 ± .009 | .75 ± .009 | .76 ± .008 | **.77** ± .008 | .76 ± .008 | .70 ± .009 | .70 ± .009 | **.72** ± .009 | **.72** ± .009 | .71 ± .009 |
| SNR | | | | | Gaussian Noise | | | | | |
| 29 low noise | .74 ± .009 | .75 ± .008 | .76 ± .008 | **.77** ± .008 | .76 ± .008 | .70 ± .009 | .69 ± .009 | **.71** ± .009 | **.71** ± .009 | .70 ± .009 |
| 15 medium noise | .59 ± .010 | .61 ± .010 | .66 ± .009 | .62 ± .010 | **.68** ± .009 | .54 ± .010 | .56 ± .010 | .60 ± .010 | .53 ± .010 | **.62** ± .010 |
| 9 high noise | .22 ± .008 | .22 ± .008 | .28 ± .009 | .20 ± .008 | **.37** ± .009 | .18 ± .008 | .20 ± .008 | .23 ± .008 | .16 ± .007 | **.27** ± .009 |

Table 3: Performance comparison using average test accuracy of different models under FGSM Adversarial Attacks for MNIST.

| Model | Det | MC Drop | BCNN | exVI | enVI | unVI | Ensemble | BaSIS 100 | BaSIS 500 | BaSIS 1000 |
|---|---|---|---|---|---|---|---|---|---|---|
| Noise Free | .96 ± .004 | .96 ± .004 | **.97** ± .003 | **.97** ± .003 | **.97** ± .003 | **.97** ± .003 | .96 ± .004 | **.97** ± .003 | **.97** ± .003 | **.97** ± .003 |
| SNR | FGSM Targeted Attack - Fooling Class: Digit 3 | | | | | | | | | |
| 16 low noise | .85 ± .007 | .90 ± .006 | .79 ± .008 | .92 ± .005 | .90 ± .006 | .90 ± .006 | .85 ± .007 | .94 ± .005 | **.95** ± .004 | **.95** ± .004 |
| 5 high noise | .14 ± .007 | .23 ± .008 | .30 ± .009 | .85 ± .007 | .84 ± .007 | .85 ± .007 | .25 ± .008 | .86 ± .007 | **.87** ± .007 | **.87** ± .007 |
| | FGSM Untargeted Attack | | | | | | | | | |
| 15 low noise | .85 ± .007 | .90 ± .006 | .75 ± .008 | .73 ± .005 | .70 ± .006 | .72 ± .006 | .87 ± .007 | .94 ± .005 | **.95** ± .004 | **.95** ± .004 |
| 3 high noise | .14 ± .007 | .22 ± .008 | .10 ± .006 | .41 ± .010 | .42 ± .010 | .42 ± .010 | .20 ± .008 | .86 ± .007 | .86 ± .007 | **.87** ± .007 |

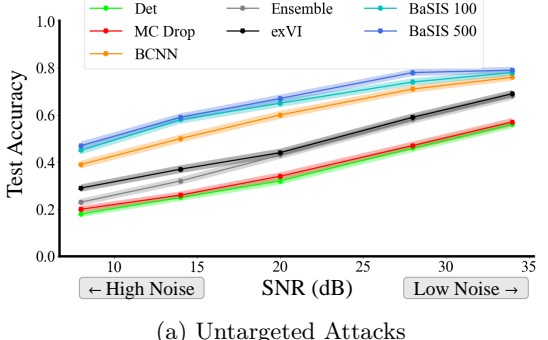
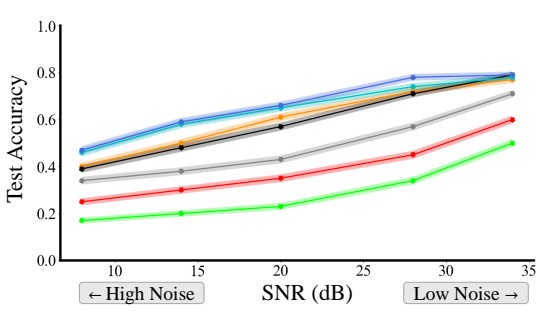

(a) Untargeted Attacks

(b) Targeted Attacks with fooling class label 3

Figure 2: Performance of different models under various levels of FGSM Adversarial noise applied to CIFAR-10 test data. The $y$-axis displays the average test accuracy and $x$-axis represents the strength of the attack measured using Signal to Noise Ratio (SNR) in units of dB. We note the performance benefits of using BaSIS-Net at low SNR values (high noise) for both attack types.

## 5 Results

We present the results of our study, which demonstrate the superior performance of BaSIS-Net in comparison to other state-of-the-art Bayesian and approximate-Bayesian methods, on two classical benchmark datasets, namely MNIST digits and CIFAR-10, and two medical imaging datasets, Malaria classification, and Hippocampus segmentation datasets (Deng, 2012; Krizhevsky & Hinton, 2009; Simpson et al., 2019; Rajaraman et al., 2018). We evaluate BaSIS-Net against other state-of-the-art Bayesian and approximate-Bayesian approaches including BCNN, MC Drop, Ensemble, and three variants of VDP (exVI, enVI, unVI). For reference, we include results for a deterministic network (Det).

### 5.1 Performance Analysis Under Gaussian Noise

Figure 1 and Tables 1 and 2 display the average test accuracy of various models when Gaussian noise is added to the test dataset samples. Figure 1a shows the average test accuracy plotted against various levels of Gaussian noise, measured using SNR, added to the test dataset of MNIST. The performance of Det, MC Drop, BCNN, exVI, enVI, unVI, Ensemble, and BaSIS-Net are shown. In Figure 1b, the same plot is presented for the Malaria dataset with Det, MC Drop, BCNN, Ensemble, and BaSIS-Net. Table 1 reports the performance of Det, MC Drop, BCNN, exVI, Ensemble, and BaSIS-Net for three levels of Gaussian noise added to the CIFAR-10 test dataset. Similarly, Table 2 provides the same results for the Hippocampus test dataset.

Table 4: Performance comparison using average test accuracy of different models under PGD Adversarial Attacks for CIFAR-10.

| Model | Det | MC Drop | BCNN | exVI | Ensemble | BaSIS 100 | BaSIS 500 |
|---|---|---|---|---|---|---|---|
| Noise Free | .84 ± .007 | .84 ± .007 | .83 ± .007 | **.86** ± .007 | .84 ± .007 | .84 ± .007 | .84 ± .007 |
| SNR | colspan: PGD Targeted Attack; Fooling class: Label 8; Step-size: 1; Max-iter: 10 | | | | | | |
| 20 low noise | .45 ± .010 | .49 ± .010 | .44 ± .010 | **.58** ± .010 | **.58** ± .010 | .57 ± .010 | **.58** ± .010 |
| 14 high noise | .37 ± .009 | .38 ± .010 | .36 ± .009 | .46 ± .010 | .46 ± .010 | .46 ± .010 | **.47** ± .010 |
| | colspan: PGD Targeted Attack; Fooling class: Label 3; Step-size: 1; Max-iter: 40 | | | | | | |
| 20 low noise | .44 ± .010 | .49 ± .010 | .44 ± .010 | .54 ± .010 | **.59** ± .010 | .58 ± .010 | .58 ± .010 |
| 14 high noise | .37 ± .009 | .45 ± .010 | .37 ± .009 | .45 ± .010 | .46 ± .010 | .46 ± .010 | **.47** ± .010 |
| | colspan: PGD Untargeted Attack; Step-size: 1; Max-iter: 10 | | | | | | |
| 20 low noise | .12 ± .006 | .18 ± .008 | .53 ± .010 | .53 ± .010 | .38 ± .010 | .57 ± .010 | **.58** ± .010 |
| 14 high noise | .10 ± .006 | .15 ± .007 | .44 ± .010 | .40 ± .010 | .29 ± .009 | **.46** ± .010 | **.46** ± .010 |
| | colspan: PGD Untargeted Attack; Step-size: 1; Max-iter: 40 | | | | | | |
| 20 low noise | .11 ± .006 | .17 ± .007 | .54 ± .010 | .54 ± .010 | .37 ± .009 | **.58** ± .010 | **.58** ± .010 |
| 14 high noise | .10 ± .006 | .14 ± .007 | .43 ± .010 | .41 ± .010 | .27 ± .009 | **.46** ± .010 | **.46** ± .010 |

Table 5: Performance comparison using average test accuracy of different models under Adversarial Attacks for Malaria Dataset

| Model | Det | MC Drop | BCNN | Ensemble | BaSIS 100 |
|---|---|---|---|---|---|
| Noise Free | **.94** ± .008 | **.95** ± .007 | .94 ± .008 | .94 ± .008 | .93 ± .009 |
| SNR | colspan: FGSM Untargeted Attack | | | | |
| 33 low noise | .50 ± .017 | .55 ± .017 | .50 ± .017 | .70 ± .015 | **.87** ± .011 |
| 25 medium noise | .39 ± .016 | .48 ± .017 | .44 ± .017 | .60 ± .016 | **.83** ± .013 |
| 13 high noise | .36 ± .016 | .41 ± .016 | .41 ± .016 | .50 ± .017 | **.72** ± .015 |
| | colspan: PGD Untargeted Attack Step-size: 1; Max-iter: 10 | | | | |
| 33 low noise | .40 ± .016 | .45 ± .017 | .54 ± .017 | .60 ± .016 | **.81** ± .013 |
| 21 medium noise | .37 ± .016 | .29 ± .015 | .52 ± .017 | .70 ± .015 | **.87** ± .011 |
| 13 high noise | .34 ± .016 | .20 ± .013 | .38 ± .016 | .47 ± .017 | **.57** ± .017 |

## 5.2 Performance Analysis Under Adversarial Noise

Tables 3, 4, 5, 6 and Fig. 2 present a comprehensive evaluation of the robustness of several models, including Det, MC Drop, BCNN, exVI, enVI, unVI, Ensemble, and BaSIS-Net, under various adversarial attacks. Table 3 shows the performance of these models against FGSM attacks on MNIST. Similarly, Table 4 reports results for targeted and untargeted PGD attacks on CIFAR-10 for Det, MC Drop, BCNN, exVI, Ensemble and BaSIS-Net. Fig. 2 depicts the performance of these models under FGSM attacks on CIFAR-10. Table 5 provides the performance comparison for untargeted FGSM and PGD attacks on the Malaria dataset for Det, MC Drop, BCNN, Ensemble, and BaSIS-Net. Finally, Table 6 reports the results of targeted PGD attacks on the Hippocampus dataset for Det, MC Drop, exVI, Ensemble, and BaSIS-Net.

Table 6: Performance comparison using average DSC of different models under PGD Adversarial Attacks for Hippocampus Dataset.

| Model | Anterior | | | | | Posterior | | | | |
|---|---|---|---|---|---|---|---|---|---|---|
| | Det | MC Drop | exVI | Ensemble | BaSIS 100 | Det | MC Drop | exVI | Ensemble | BaSIS 100 |
| Noise Free | .74 ± .015 | .75 ± .014 | .76 ± .014 | **.77** ± .014 | .76 ± .014 | .70 ± .015 | .70 ± .015 | **.72** ± .015 | **.72** ± .015 | .71 ± .015 |
| SNR | colspan: PGD Targeted Attack. Fooling class: Label 2; Step-size: 1; Max-iter: 5 | | | | | | | | | |
| 28 low noise | .66 ± .016 | .61 ± .016 | .61 ± .016 | **.67** ± .016 | .66 ± .016 | .55 ± .017 | .66 ± .016 | .66 ± .016 | .63 ± .016 | **.67** ± .016 |
| 14 medium noise | .18 ± .013 | .15 ± .012 | .23 ± .014 | .21 ± .014 | **.24** ± .009 | .08 ± .015 | .31 ± .015 | .28 ± .015 | .26 ± .016 | **.34** ± .021 |
| 8 high noise | 0 | 0 | .02 ± .005 | .02 ± .005 | **.05** ± .007 | .01 ± .003 | .03 ± .006 | .03 ± .006 | .02 ± .005 | **.06** ± .008 |

### 5.3 Predictive Uncertainty

The model uncertainty is captured by the variance or the second central moment of the predictive probability density function (pdf). The "predictive uncertainty" is defined as the (sample) variance of the predictive distribution. The predicted class corresponds to the mean of the predictive distribution. We investigate the behavior of the predictive variance when perturbations, such as Gaussian noise or adversarial attacks, are added to the test datasets. We present the average predictive variance under various levels of (1) Gaussian noise, (2) untargeted FSGM attacks and (3) targeted FGSM attacks on MNIST and CIFAR-10 datasets in Figs. 3 and 4. The average predictive variance values are normalized between 0 and 1. We show the results for BaSIS-Net models with 100 and 500 particles in Figs. 3 and 4, respectively, and compare with other Bayesian approaches. For a characterization of the calibrateness of the model, we computed the Brier score. Fig. 5 shows the Brier score when various levels of (1) Gaussian noise, (2) untargeted FSGM attacks and (3) targeted FGSM attacks are applied to MNIST test data. We note that at high levels of noise (both Gaussian and adversarial), the model test accuracy drops (as expected), and the Brier score increases.

For the segmentation task, we introduce pixel-level "Uncertainty Maps" that associate predictive variance to each pixel in the model output. We demonstrate a representative case from the Hippocampus dataset in Fig. 6. Each row refers to a method: (1) Basis 100, (2) MC Drop, (3) exVI, (4) Ensemble. From left to right: (a) input image, (b) ground-truth segmentation, (c) noise-free segmentation prediction with (d) associated uncertainty map, (e) additive Gaussian noise prediction and (f) associated uncertainty map, and (g) a targeted adversarial attack prediction and (h) associated uncertainty map. We normalize all uncertainty maps between 0 and 1 for better visual comparison. For the Gaussian noise case, we use a perturbation with SNR $\approx 15$, and for the adversarial attack, we use SNR $\approx 20$. The solid arrows in Fig. 6 point to pixels that are incorrectly classified in the prediction segmentation maps and their corresponding spots on the uncertainty maps. We note that the pixels in the uncertainty maps that correspond to the incorrect segmentation reflect low confidence (higher variance values). The dashed arrows point to regions with inconsistent uncertainty estimates, i.e., high variance with no incorrect predictions.

### 5.4 Predictive Distribution

We investigate the effect of perturbations on the predictive pdf of BaSIS-Net. Specifically, we examine the changes in the pdf when perturbations, such as Gaussian noise and adversarial attacks, are applied to the input test samples. To this end, we randomly select two samples from the MNIST test dataset and feed noise-free and perturbed images to BaSIS-500 model. We plot the number of individual particles on the $y-$axis against softmax scores on the $x-$axis. We consider two levels of additive Gaussian noise and two different adversarial attacks, with perturbation strengths quantified by signal-to-noise ratios (SNRs) of $\approx 13$ and 3 for Gaussian noise, and $\approx 15$ and 5 for FGSM adversarial attacks with a fooling class label of 3. These perturbations affect the pdf and modify the distribution of individual particles as shown in Fig. 7.

## 6 Discussion

Our objective was to design resilient models capable of producing predictions while also providing confidence estimates. To achieve this, we utilized the Sequential Importance Sampling (SIS) method as our primary tool to build DL models that generate samples from the approximate predictive distribution. We computed the second moment of this distribution and referred to it as the predictive uncertainty. To evaluate the robustness of our models, we exposed them to different types of noise and compared their performance with that of other cutting-edge Bayesian and approximate Bayesian models.

### 6.1 Increased Robustness

Based on our findings presented in Tables 1–6 and Figs. 1a, 1b, and 2, we can observe that our proposed BaSIS models exhibit enhanced robustness to noise and adversarial attacks. Specifically, at lower SNRs (high noise levels), the BaSIS models achieve a higher prediction accuracy as compared to other models. It is noteworthy that our approach does not require sampling at inference time. The BaSIS framework is comparable to VDP at low noise levels and outperforms it at higher noise levels and adversarial attacks. We

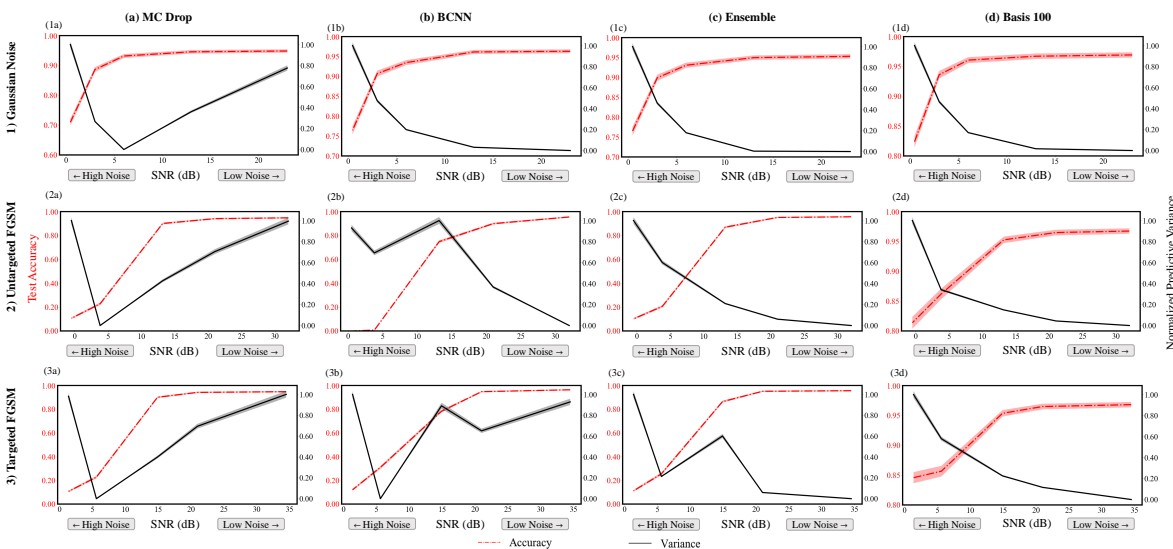

Figure 3: Average test accuracy (red color line) and average predictive variance (black color line) are presented at various levels of (1) Gaussian noise, (2) Untargeted FGSM adversarial noise, and (3) Targeted FGSM adversarial noise applied to MNIST test dataset. The $x-$axis reports Signal to Noise Ratio (SNR), measured in dB. We note that at high levels of noise (both Gaussian and adversarial), the Basis model test accuracy drops (as expected), and predictive variance increases (d). Other approaches do not offer the same decreasing monotonic behavior (a - c). We use the predictive variance to quantify BaSIS-Net's confidence in its output.

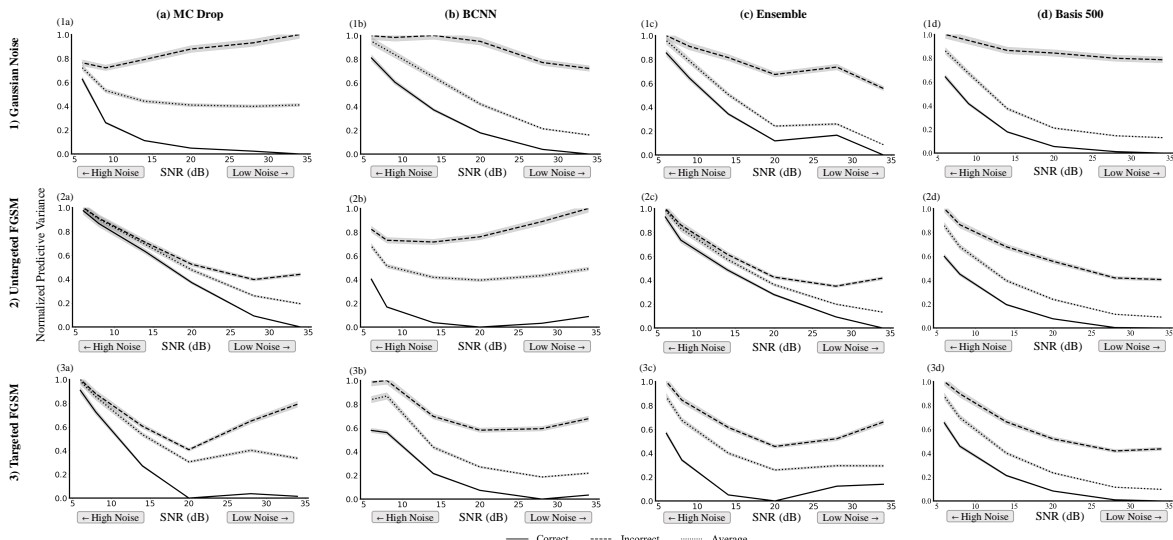

Figure 4: Average predictive variance ($y$-axis) are presented at various levels of (1) Gaussian noise, (2) Untargeted FGSM adversarial noise, and (3) Targeted FGSM adversarial noise applied to CIFAR-10 test data. $x-$axis reports Signal to Noise Ratio (SNR), measured in dB. Dotted lines show the average predictive variance across all test inputs, solid lines represent correctly classified, and dashed lines incorrectly classified inputs. Basis-Net (d) shows a monotonic increasing behavior at decreasing SNR while exhibiting lower confidence (or higher uncertainty) when their output is incorrect unlike other compared approaches (Nguyen et al., 2015).

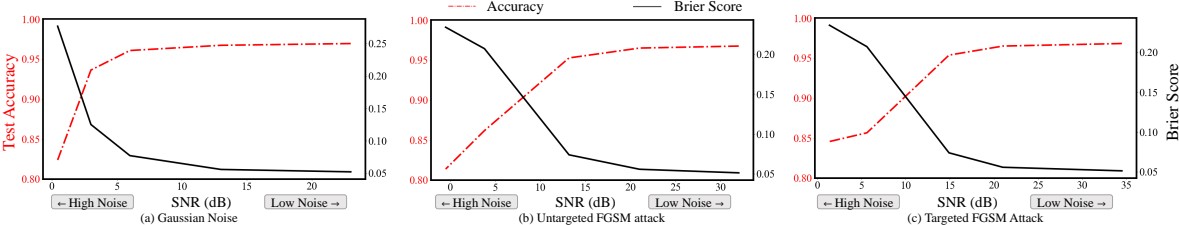

Figure 5: BaSIS 100 average test accuracy and Brier values are presented at various levels of Gaussian and adversarial noise. Left $y$-axis (red color line) represents average test accuracy and right $y$-axis (black color line) represents Brier scores calculated for (a) Gaussian noise, (b) Untargeted FGSM adversarial noise, and (c) Targeted FGSM adversarial noise applied to MNIST test dataset. The $x$−axis reports Signal to Noise Ratio (SNR), measured in dB. We note that at high levels of noise (both Gaussian and adversarial), the model test accuracy drops (as expected), and the Brier score increases.

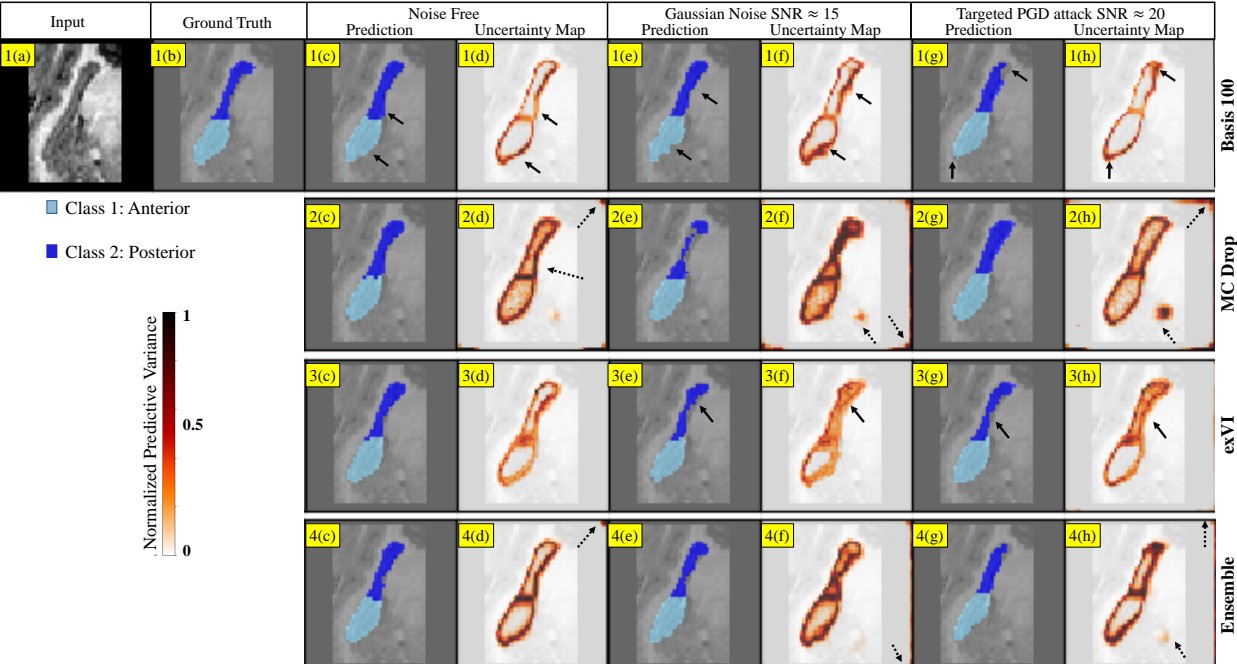

Figure 6: Representative segmentation results for the Hippocampus dataset. Each row refers to a method: (1) Basis 100, (2) MC Drop, (3) exVI, (4) Ensemble. From left to right: (a) input image, (b) ground-truth segmentation, (c) noise-free segmentation prediction with (d) associated uncertainty map, (e) additive Gaussian noise prediction and (f) associated uncertainty map, and (g) a targeted adversarial attack prediction and (h) associated uncertainty map. The solid arrows point to regions incorrectly classified by the models that correspond to low confidence (higher variance values). The dashed arrows point to regions with inconsistent uncertainty estimates, i.e., high variance with no incorrect predictions.

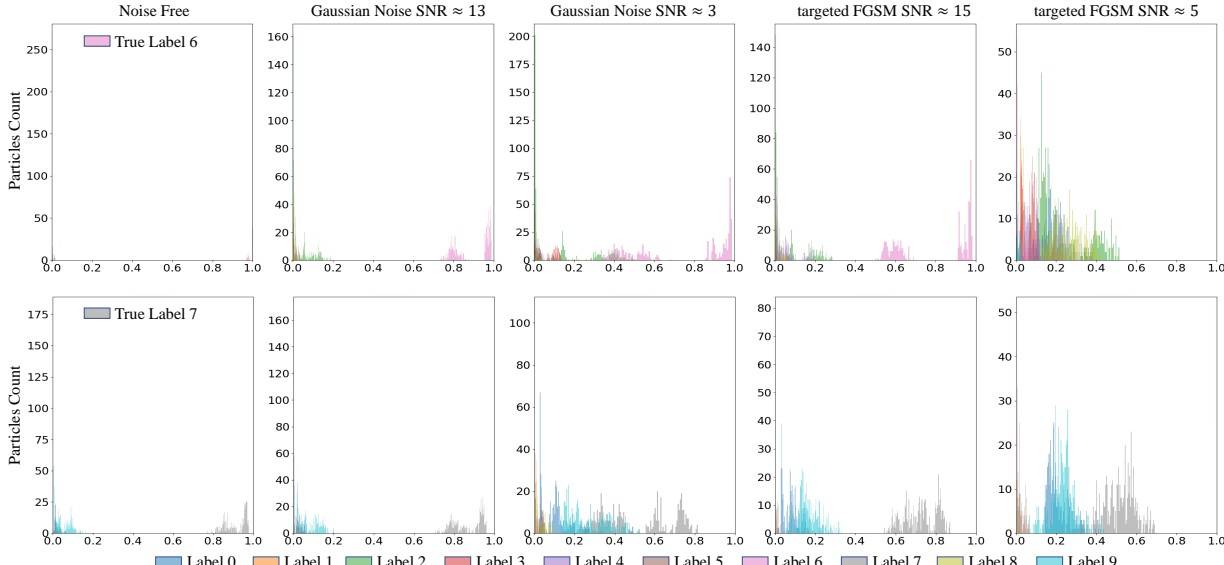

Figure 7: BaSIS 500 predictive distributions for two randomly selected images from MNIST test dataset. For each example, we show (left to right) the prediction for the noise-free case, two Gaussian noise cases with SNRs $\approx 13$, $3$, and two FGSM attacks with fooling class digit 3 and SNRs $\approx 15$, $5$. $x-$axes show the particles' predictions, i.e., softmax scores, while $y-$axes represent particles' count. We observe that the variances of the predictive distributions increase with increasing noise levels, reflecting higher uncertainty (lower confidence) in the predictions at high noise levels. As we increase the perturbations, the spread and the number of modes of the distributions increase.

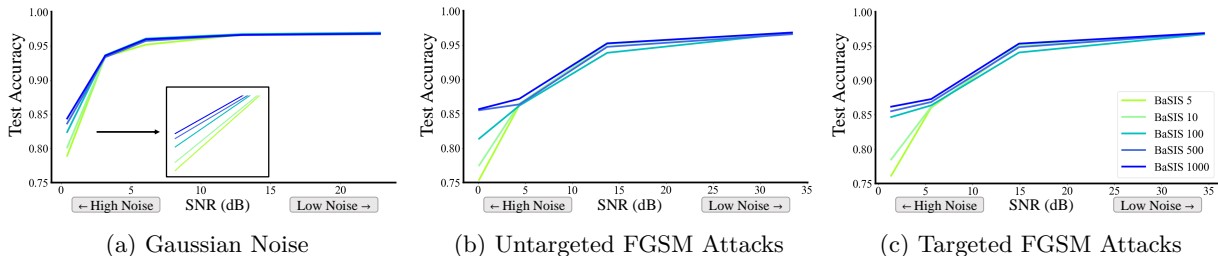

Figure 8: The effect of the number of particles $N$ on BaSIS-Net performance is presented at various levels of Gaussian and adversarial noise. $y$-axis shows the average test accuracy calculated for (a) Gaussian noise, (b) Untargeted FGSM adversarial noise, and (c) Targeted FGSM adversarial noise applied to MNIST test dataset. $x-$axis reports Signal-to-Noise Ratio (SNR) values, measured in units of decibels (dB). For Gaussian noise, the box shows the lower SNR values zoomed. We note that at high levels of noise (both Gaussian and adversarial), BaSIS models with a higher number of particles $N$ have higher average test accuracy.

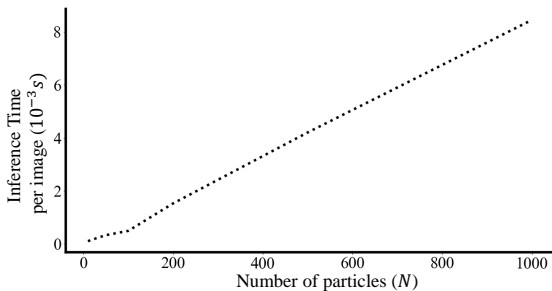

Figure 9: BaSIS computational complexity at inference time for MNIST data. $y-$axis shows the inference time per image $(10^{-3}s)$ and $x-$axis reports the number of particles $N$ used in BaSIS-Net.

Table 7: Training Time for all Datasets and Models

| per epoch $(min)$ | Det | MC Drop | BCNN | exVI | enVI | unVI | ensemble | BaSIS 100 | BaSIS 500 | BaSIS 1000 |
|---|---|---|---|---|---|---|---|---|---|---|
| MNIST | 1.36 | 1.39 | 2.53 | 31.11 | 76.41 | 79.94 | $1.36 \cdot N_2$ | 4.50 | 19.39 | 26.21 |
| CIFAR-10 | 1.39 | 1.43 | 2.45 | 16.43 | – | – | $1.39 \cdot N_2$ | 13.88 | 37.26 | – |
| Malaria | 9.17 | 10.44 | 15.47 | – | – | – | $9.17 \cdot N_2$ | 58.90 | – | – |
| Hippocampus | 10.55 | 11.36 | – | 34.69 | – | – | $10.55 \cdot N_2$ | 32.71 | – | – |

$N_2$ is the number of networks in the ensemble.

postulate that propagating higher-order moments, thanks to the use of particles, and freeing the posterior from functional constraints contribute to the increased performance and robustness to noise and adversarial attacks exhibited by BaSIS models.

## 6.2 The Effect of Number of Particles

The impact of the number of particles $N$ on the robustness of BaSIS models was investigated. The results presented in Fig. 8 indicate that increasing the number of particles leads to improved performance, particularly at lower SNR values (i.e., high noise levels). This observation is consistent with the mathematical convergence guarantees for SIS and PFs. Specifically, under mild conditions, PFs are guaranteed to converge asymptotically to the true posterior pdf (almost surely and in the mean square error sense) as $N$ increases. While our empirical results demonstrate promising classification performance, the direct correlation to the quality of PF predictive inference, given the methodological modifications we've employed in evaluating likelihood/loss instead of a per-particle basis (step 5 in Algorithm 1), warrants further investigation (Izmailov et al., 2021).

## 6.3 Evaluation of the Predictive Uncertainty

Our experimental findings indicate that the predictive uncertainty estimated by BaSIS models exhibits a monotonic trend. Specifically, as the levels of noise and adversarial attacks increase, the average predictive variance also increases. This behavior is illustrated in Fig. 3, where we observe that the Basis average predictive variance decreases monotonically with increasing SNR. In other words, BaSIS models tend to exhibit reduced accuracy and higher uncertainty under higher levels of noise. On the contrary, the other approaches do not show the same ability. Furthermore, Fig. 4 reveals that: i) the average predictive variance (dotted line) increases monotonically for decreasing SNRs, and ii) the predictive variance of incorrectly classified samples (dashed lines) is consistently higher than correctly classified ones (solid lines). This suggests

Table 8: Inference Time for all Datasets and Models

| per image $(s^{-3})$ | Det | MC Drop | BCNN | exVI | enVI | unVI | ensemble | BaSIS 100 | BaSIS 500 | BaSIS 1000 |
|---|---|---|---|---|---|---|---|---|---|---|
| MNIST | 0.04 | $0.05 \cdot N_1$ | $0.05 \cdot N_1$ | 3.04 | 8.13 | 8.80 | $0.04 \cdot N_2$ | 0.49 | 4.20 | 8.45 |
| CIFAR-10 | 0.13 | $0.13 \cdot N_1$ | $0.16 \cdot N_1$ | 4.45 | – | – | $0.13 \cdot N_2$ | 2.47 | 12.16 | – |
| Malaria | 0.07 | $0.07 \cdot N_1$ | $0.09 \cdot N_1$ | – | – | – | $0.08 \cdot N_2$ | 0.83 | – | – |
| Hippocampus | 0.75 | $0.75 \cdot N_1$ | – | 16.14 | – | – | $0.75 \cdot N_2$ | 12.02 | – | – |

$N_1$ represents the number of samples, i.e., forward runs through the network, at inference time.
$N_2$ is the number of networks in the ensemble.

that BaSIS models are capable of detecting perturbations in the input data and distinguishing between correct and incorrect predictions (Carannante et al., 2022; McCrindle et al., 2021).

In addition, Fig. 6 shows that BaSIS models tend to be uncertain near the borders of the structures of interest and for misclassified pixels. The black arrows placed on the predictions near the misclassified pixels correspond to regions of high uncertainty on the uncertainty maps. This *self-aware* behavior could increase user trust in the model, as a human user would be able to determine whether the BaSIS model is performing reliably. Overall, our results indicate that BaSIS models are capable of producing predictions with high accuracy and reliable estimates of predictive uncertainty, even in the presence of noise and adversarial attacks.

### 6.4 Computational Complexity

The computational demand of SIS and PFs algorithms can be substantial, particularly when the number of particles, $N$, increases. In particular, if the model has $d$ parameters, PF requires storage $d \cdot N$ particles. For instance, in the MNIST architecture we used, a deterministic model and MC Drop would need 183.13 KB of storage using $4 \cdot d/1024$ with $d = 46880$. Bayesian methods, such as BCNN or VDP, would require roughly twice that amount because they have twice the number of parameters, i.e., means and variances. The storage requirement of Ensemble and Basis scales with the number of models and particles, respectively. This can lead to a trade-off between the number of particles and the performance of BaSIS.

The interplay between the performance of BaSIS model, the number of particles $N$, and computational demands is illustrated in Figs. 1a and 8, as well as Tables 8 and 7. The average inference time per image is plotted against the number of particles $N$ of BaSIS-Net for MNIST in Fig. 8. It is observed that the test time approximately scales linearly with the number of particles $N$. We present a comparison of the training time and inference time for all approaches in Tables 7 and 8, respectively. Depending on the number of particles $N$ and the specific dataset, the computational complexity of training BaSIS may be lower compared to VDP frameworks (exVI, enVI, unVI), which propagate a Taylor-series approximation of the mean and variance of the variational distribution. In general, other approaches, including MC Drop, BCNN, and Ensemble, are less computationally demanding. This difference arises from these models using point-estimate methods, where a single sample is propagated through the model layers. However, when employing these models as approximate Bayesian methods, multiple samples need to be considered at inference time to estimate uncertainty. For instance, the inference time of Basis with 10 particles is 0.10 $(10^{-3}s)$, whereas an ensemble of 10 models would require $0.04 \cdot 10$ $(10^{-3}s)$."

Training is conducted offline, which allows for flexibility in managing computational resources. Notably, Table 7 shows that the inference time for BaSIS with 100 particles is comparable to that of MC Dropout and ensemble methods involving approximately 10 to 20 models. This equivalency in inference time efficiency underscores an important trade-off: the computational complexity incurred during BaSIS's training phase is offset by several key advantages. Firstly, BaSIS demonstrates significant improved robustness in noisy and adversarial conditions, showcasing its resilience and reliability. Secondly, its design facilitates self-assessment capabilities through the utilization of the second moment of the predictive distribution, providing a deeper insight into the model's uncertainty. Lastly, BaSIS opens the door to exploring the impact and effects of higher-order moments on accuracy, robustness, and self-assessment. Although the exploration of higher-order moments falls beyond the scope of this paper, it highlights the promising potential of BaSIS for pioneering research in this domain. This upfront computational investment during training could be a worthwhile exchange for the gains achieved in model performance and analytical depth.

## 7 Conclusion

In this work, we proposed a novel framework, BaSIS-Net, for learning posterior and predictive probability distribution functions through a set of weighted samples or particles. The proposed BaSIS-Net is more robust to noise and adversarial attacks than other approaches, as demonstrated through extensive simulations. The SIS framework enables the approximation of the full predictive distribution, which provides a better picture and assists in evaluating the reliability of the model's predictions. We presented the monotonic property of

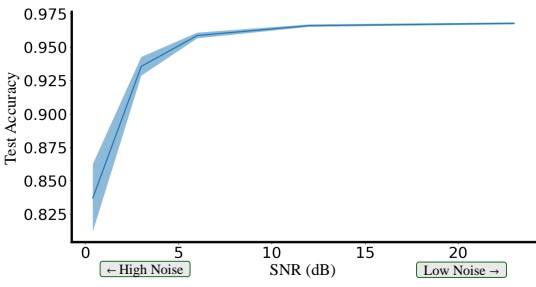

Figure 10: Average BaSIS test accuracy along with the standard deviation (shaded region) obtained with 10 random initializations. We consider different levels Gaussian noise applied to the MNIST test data measured as SNRs in dB.

the uncertainty estimated by BaSIS-Net, where the average uncertainty increases with decreasing SNR (or increasing noise). Additionally, we noted that the average predictive variance associated with the incorrect predictions is higher than the correctly classified ones. This information can be used to introduce self-awareness in BaSIS models, making these more suitable for deployment in critical application areas, such as medical diagnosis. Overall, our proposed BaSIS-Net framework provides a promising avenue for developing more robust and reliable machine learning models.

## Acknowledgment

This work was supported by the National Science Foundation awards ECCS-1903466, OAC-2234836, and PFI-2234468. We acknowledge the support from the UK's Engineering and Physical Sciences Research Council (EPSRC) through EP/T013265/1 project NSF-EPSRC: "ShiRAS. Towards Safe and Reliable Autonomy in Sensor Driven Systems", the support for ShiRAS by the USA National Science Foundation under Grant NSF ECCS 1903466.

Table 9: Test Accuracy for different Initializations of Basis for MNIST data

| | $Basis_1$ | $Basis_2$ | $Basis_3$ | $Basis_4$ | $Basis_5$ | $Basis_6$ | $Basis_7$ | $Basis_8$ | $Basis_9$ | $Basis_{10}$ | Average | Std |
|---|---|---|---|---|---|---|---|---|---|---|---|---|
| Test Accuracy | 0.967 | 0.968 | 0.966 | 0.9677 | 0.968 | 0.968 | 0.967 | 0.968 | 0.968 | 0.968 | 0.968 | 0.0007 |

Table 10: Test Accuracy for different $\sigma_\eta$ of Basis for MNIST data

| $\sigma_\eta$ | 0.1 | 0.01 | 0.005 | 0.001 | 0.0001 | 0.00001 |
|---|---|---|---|---|---|---|
| Test Accuracy | $0.201 \pm .008$ | $0.776 \pm .008$ | $0.961 \pm .005$ | $0.967 \pm .005$ | $0.968 \pm .005$ | $0.968 \pm .005$ |

Table 11: Test Accuracy for different $\sigma_\eta$ of Basis for CIFAR-10 data

| $\sigma_\eta$ | 0.0005 | 0.0001 | 0.00001 |
|---|---|---|---|
| Test Accuracy | $0.225 \pm .008$ | $0.588 \pm .009$ | $0.841 \pm .007$ |

## A  Appendix

In this study, we investigate the robustness of the proposed BaSIS-Net by initializing it randomly multiple times. We find that different seeds lead to roughly equivalent accuracy, with consistent results observed across various random initializations. The test accuracies on the noise-free MNIST test data for the BaSIS-100 model obtained with 10 random initializations are reported in Table 9. The average test accuracy is 0.97 with standard deviation of 0.0007. Furthermore, we replicated this experiment across different levels of

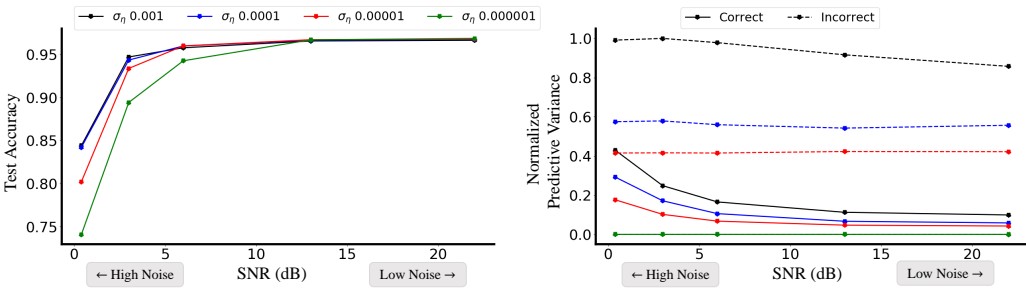

Figure 11: Impact of transition noise $\sigma_\eta$ on test accuracy and predictive variance in MNIST test data. Average BaSIS test accuracy versus Signal-to-Noise-Ratio (SNR) in dB (left). Average normalized predictive variance versus SNR (right). The different $\sigma_\eta$ selections are represented by the various colors. For the normalized predictive variance, solid lines represent correct classifications, and dashed lines represent incorrect ones. Note that for the lowest $\sigma_\eta$ in green, there is no difference in variance between correct and incorrect classification.

Gaussian noise. We observed that the accuracies of the different BaSIS-Net models only showed variation within the second decimal place at the lowest SNR of $\approx 0$, further validating the reliability of Basis-Net under noisy conditions as well (Fig. 10).

We evaluated the impact of the transition noise $\sigma_\eta$ on the accuracy of BaSIS-Net (i.e., the mean of the predictive distribution). Tables 10 and 11 present the accuracy of BaSIS-Net versus $\sigma_\eta$ on the MNIST and CIFAR-10 dataset as a proof of concept. Observe that the model accuracy remains robust across a large range of sigma values, specifically for $\sigma_\eta < 0.005$ (Table 10). Accuracy begins to decline beyond this value, and the model diverges for $\sigma_\eta = 0.01$.

We evaluated the impact of the transition noise parameter $\sigma_\eta$ on the robustness of BaSIS-Net to noisy conditions and assessed the predictive uncertainty of the resulting models. Our findings indicate that smaller $\sigma_\eta$ values are associated with higher accuracy on clean noiseless data. However, at higher noise levels, models with larger $\sigma_\eta$ exhibit more robust performance. Additionally, for low $\sigma_\eta$ values, the uncertainty estimates are not as informative as in models trained with larger values.

In Fig. 11, we illustrate how test accuracy and variance information vary with different $\sigma_\eta$ selections under noisy conditions applied to the MNIST test data. We observe that models trained with a small $\sigma_\eta$ are less robust at low SNR, and their variance is unable to distinguish between correctly and incorrectly classified inputs. This undermines one of the key features of our BaSIS-Net framework, which is the ability to provide meaningful uncertainty estimates that differentiate between correct and incorrect classifications.

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
