**BaSIS-Net: From Point Estimate to Predictive Distribution in Neural Networks - A Bayesian Sequential Importance Sampling Framework**

The authors would like to thank the reviewers for their valuable comments and constructive suggestions.

**REVIEWER X46h**

**The figures and tables in the paper lack uncertainty intervals and, furthermore, information about how many seeds the authors averaged in the results.**

**Answer**: First, please allow us to clarify the essence of our proposed PF approach. PF is not a classic ensembling approach, as widely adopted in ML, but a technique to learn (or track) the posterior distribution of the parameters given the (training) data. As such, the output of the Basis-Net model is the entire predictive distribution of the input (as in Eq. (1)) rather than a point-estimate. In the PF framework, a distribution is represented by its discrete approximation as in Eq.(5), i.e., the set of samples (also called particles) and their corresponding ratios (i.e., mass probabilities). In particular, the particles represent estimates of the model parameters. The model decision (e.g., classification) is given by the mean of the predictive distribution whereas the model uncertainty is given by the variance of this distribution.

In classical ensembling approaches, "seeds" refer to various initializations of the model, requiring the training of multiple models with different seeds to estimate uncertainty as the sample variance of decisions made by these models. In contrast, our PF approach provides, at test time, the entire predictive distribution of the input. Both the model point decision and variance can be directly derived from this predictive distribution. In particular, Basis-Net eliminates the need for Monte Carlo simulations or seeds at test time. This is one of the main key advantages of the proposed approach.

Yet, the reviewer's point regarding the use of seeds to confirm model reliability (i.e., different seeds lead to roughly equivalent accuracies) is acknowledged. First, we would like to clarify that we did not use a fixed seed. All experiments reported in this paper utilized random seeds for every model and dataset. To directly address the reviewer's comment, we trained ten Basis-Net models, each with a different seed. The results from the MNIST dataset show that the accuracy achieved by these models is consistent to the third decimal place (Table 9 in Appendix) with an average of 0.97 and std = 0.0007, showing that the model is reliable. Furthermore, we replicated this experiment across four levels of Gaussian noise. We observed that the accuracies of the different Basis-Net models only showed variation within the second decimal place at the lowest Signal-to-Noise Ratio (SNR) of 0.4, further validating the reliability of Basis-Net under noisy conditions as well (Figure 10 in Appendix).

Regarding the uncertainty intervals, we included the 95% confidence intervals for all experiments in the revised version of the manuscript.

**The authors do not motivate why they do not attack all models with all schemes (i.e. (un-)targeted FSGM and PGD attacks) in Section 5.2**

**Answer**: The revised version of the paper will motivate this choice by the fact that we wanted to showcase a diverse combination of attacks (Fast-Gradient Sign Method and Projected Gradient

targeted and untargeted) across a variety of datasets (MNIST, CIFAR-10, Malaria and Hippocampus). The decision was guided by the understanding that the strength of an attack depends on multiple factors, including the inherent characteristics of the dataset. Furthermore, we conducted experiments using different levels of attacks, varying from higher to lower power (e.g., measured by SNR from 33dB to 8dB in Fig. 2). Our rationale behind this approach was to assess the model's resilience across a spectrum of attack intensities. We argue that a high-level PGD attack can potentially approximate the impact of other potential attack schemes. Therefore, by varying the intensity of PGD attacks, we aimed to capture a wide range of adversarial scenarios and provide a comprehensive analysis of the model's robustness). Yet, we did consider complex datasets and tasks (i.e., Hippocampus segmentation task) under an advanced attack, i.e., PGD (see Tables 2 and 6), which would be a harsh test for our algorithm. Finally, to maintain brevity and avoid an overly lengthy paper, we cannot display every dataset under every attack model. For some datasets, we made a deliberate choice; for example, considering the binary nature of the Malaria dataset, which categorizes cells as either parasitized or uninfected, we opted for untargeted attacks to encompass broader adversarial conditions than merely focusing on individual class targeting.

Following the reviewer's suggestion, we added a more detailed discussion when presenting our experiments. Please find below a relevant portion from Section 4.3 in the revised version of the manuscript:

"The study aims to showcase various attacks (FGSM and PGD targeted and untargeted) across diverse datasets (MNIST, CIFAR-10, Malaria, and Hippocampus), recognizing that attack effectiveness depends on the power of attack as well as inherent dataset characteristics. Experiments varied the attack intensity (measured by SNR) from high to low to comprehensively assess model resilience. We systematically explored a variety of attack scenarios by employing a mix of targeted and untargeted attacks across different datasets. This approach allowed us to showcase a broad range of adversarial situations."

**BaSIS-Net sed different noise values sigma_{eta} for different datasets and network architectures. However, the authors do not discuss the sensitivity of the method to this hyperparameter or how to choose it.**

**Answer**: In the general context of state space models for tracking probability distributions, the noise sigma_{eta}, in the transition model governs the search space for particles at the next time step. As has been well-established since the advent of the Kalman filter, the selection of $\sigma_\eta$ significantly impacts the algorithm's convergence behavior [1-2]. When $\sigma_\eta$ is excessively large, the search space becomes expansive, potentially leading to challenges in convergence given the finite number of particles propagated in practice. Conversely, when $\sigma_\eta$ is too small, the model effectively behaves almost deterministically, akin to propagating a delta function, significantly restricting the search space. When starting our experiments, for each dataset, we performed a grid-search over a set of values for $\sigma_\eta$, starting from large to small, to identify a value that guarantees convergence with a good test accuracy.

In the revised version of the manuscript, we will address this aspect more explicitly by providing insights into our approach for choosing $\sigma_\eta$ to strike a balance between exploration and

exploitation in the particle filtering process. In particular, as suggested by the reviewer, we added a subsection in 3.4 (Basis-Net Hyperparameters) to discuss it in detail. Please find below the relevant excerpt from the added discussion on this topic in the revised manuscript:

"In the general context of state space models, the noise values in the transition model govern the search space for particles at the subsequent time step. As has been well-established since the advent of the Kalman filter, the selection of σ_η impacts the algorithm's convergence behavior (Gordon et al., 1993; Park et al., 2013). When selecting the sigma parameter for the noise in the transition model of the PF, it is crucial to consider the trade-off between exploration and exploitation (or convergence). Given that in practice, we have a finite number of particles, using a large sigma encourages exploration of the state space but may hinder convergence. Conversely, a small sigma restricts the search space, reducing particle diversity and potentially compromising the effectiveness of the filtering/tracking process."

In the Appendix, we will also include the following in the revised version of the manuscript: "We evaluated the impact of the transition noise $sigma_{eta}$ on the accuracy of Basis-Net (i.e., the mean of the predictive distribution). Table 10 and 11 presents the accuracy of Basis-Net versus $sigma_{eta}$ on the MNIST and CIFAR-10 dataset as a proof of concept. Observe that the model accuracy remains robust across a large range of sigma values, specifically for σ_η <0.005 (Table 10). Accuracy begins to decline beyond this value, and the model diverges for σ_η = 0.01."

We will also add the two tables in the Appendix.

References:

[1] Park, DONG-HYEON, S. Stephanie, and R. L. Porter. "Sensitivity to Noise in Particle Filters for 2-D Tracking Algorithms." Nursing 6 (2013): 58-65.

[2] Gordon, Neil J., David J. Salmond, and Adrian FM Smith. "Novel approach to nonlinear/non-Gaussian Bayesian state estimation." IEE proceedings F (radar and signal processing). Vol. 140. No. 2. IET Digital Library, 1993.

**Although the authors present evidence for the calibrated uncertainty of their method, they do not compare the "calibratedness" with the baseline approaches.**

**Answer**: The revised manuscript includes a comparison of our method with other approaches. We argue that a useful and meaningful uncertainty measure should convey lower confidence (i.e., higher uncertainty) for low-accuracy predictions, i.e., the cases where the model is making mistakes [3-5]. Additionally, the predictive variance should also be able to capture shifts (e.g., noise or perturbations) in the input data. [4, 5]. From our comparisons in the updated Figures 3, 4, 6 (in the revised manuscript), we see that only Basis-Net behaves as described in all scenarios whereas MC-Dropout [6], Bayes-CNN [7], and Ensemble [8] exhibit uncertainty that does not demonstrate monotonic behavior with respect to SNR. In other words, the uncertainty displayed by the latter approaches can be higher in instances of more accurate predictions, which renders it "uncalibrated". In contrast, Basis-Net uncertainty remains "well-calibrated"; for example, Fig. 4 shows that Basis-Net exhibits a monotonic increasing behavior at decreasing SNR levels for noise and artifacts. For the segmentation task, Basis-Net associates higher uncertainty for regions of

incorrect segmentations or with artifacts and noise. On the other hand, other approaches generate inconsistent uncertainty estimates, i.e., they associate low uncertainty (high confidence) to incorrect predictions and/or high uncertainty (low confidence) to correctly classified regions (Fig. 6).

[3] J. Alain, F. Balsiger, and M. Reyes. "Analyzing the quality and challenges of uncertainty estimations for brain tumor segmentation." Frontiers in neuroscience 14 (2020): 501743.

[4] G. Carannante, et al. "Self-Assessment and Robust Anomaly Detection with Bayesian Deep Learning." 2022 25th International Conference on Information Fusion (FUSION). IEEE, 2022.

[5] B. McCrindle, et al. "A Radiology-focused Review of Predictive Uncertainty for AI Interpretability in Computer-assisted Segmentation." Radiology: Artificial Intelligence 3.6 (2021).

[6] A. Kendall, et al. "Bayesian segnet: Model uncertainty in deep convolutional encoder-decoder architectures for scene understanding." arXiv preprint arXiv:1511.02680, 2015.

[7] K. Shridhar, et al. "Bayesian convolutional neural networks." arXiv preprint arXiv:1806.05978, 2018.

[8] B. Lakshminarayanan, at al. "Simple and scalable predictive uncertainty estimation using deep ensembles." Adv. Neural Inf. Process. Syst., 30, 2017.

**While interesting, Figures 5 and 6 are rather qualitative and lack a quantitative aspect that enables a comparison with baseline methods.**

**Answer**: Our uncertainty estimates were always quantitative by nature. Please allow us to clarify. The model uncertainty is captured by the variance or the second central moment of the predictive probability density function (pdf). The "predictive uncertainty" is defined as the (sample) variance of the predictive distribution. The predicted class corresponds to the mean of the predictive distribution. For the segmentation task, the model outputs a predictive distribution for each pixel. We calculate the predictive variance at every pixel to create the pixel-level "uncertainty map", which assigns a predictive variance to each pixel in the output image, thereby serving as a quantitative measure of uncertainty. To visualize this without displaying a 64 x 64 matrix of values, we used a heatmap to depict these pixel-level uncertainty maps effectively.

We demonstrate a representative case from the Hippocampus dataset in Fig. 6 (in the revised manuscript). To better enable the reviewer to perform a comparison with other approaches, we modified this figure (originally only depicting our approach) by including the predictions and uncertainty maps obtained from other approaches. Each row refers to a method: (1) Basis-100, (2) MC Drop, (3) exVI, and (4) Ensemble. From left to right: (a) input image, (b) ground-truth segmentation, (c) noise-free segmentation prediction with (d) associated uncertainty map, (e) additive Gaussian noise prediction and (f) associated uncertainty map, and (g) a targeted adversarial attack prediction and (h) associated uncertainty map. We normalized all uncertainty maps between 0 and 1 for better visual comparison. In the uncertainty maps, red values represent high pixel uncertainty and white represents low pixel uncertainty. The uncertainty maps are

overlaid to the input MRI images. In this Figure, solid arrows point to regions incorrectly classified by the model but are associated with high variance values. The dashed arrows point to regions with inconsistent uncertainty estimates, i.e., high variance with accurate predictions. We show that reliable uncertainty information can guide the evaluation of the trustworthiness of the neural network predictions. The model should recognize the perturbed data and return an output (segmented image plus an associated uncertainty map) that conveys a high level of uncertainty or low confidence in the prediction [5].

In our analysis, we also focus on exploring the effect of input perturbations on the predictive pdf of the proposed BaSIS-Net. Specifically, we examine the changes in the pdf when perturbations, such as Gaussian noise and adversarial attacks, are applied to the input test samples. In Fig. 7 (in the revised manuscript), we randomly select two samples from the MNIST test dataset and feed noise-free and perturbed images to the BaSIS-500 model. We plot the predictive distribution, i.e., the number of predicted particles on the y-axis against the softmax scores on the x-axis. We consider two levels of additive Gaussian noise and two different adversarial attacks. This figure indeed shows the quality of the predictive pdf under distributional shifts scenarios. We observe that the variances of the predictive distributions increase with increasing noise levels, reflecting higher uncertainty (lower confidence) in the predictions at high noise levels. As we increase the perturbations, the spread and the number of modes of the distributions increase.

[5] B. McCrindle, et al. "A Radiology-focused Review of Predictive Uncertainty for AI Interpretability in Computer-assisted Segmentation." Radiology: Artificial Intelligence 3.6 (2021).

**The authors use a custom architecture for the CIFAR-10 task instead of more standard architectures like ResNet-18, leading to significantly worse performance on noise-free images than already shown in the context of Bayesian NNs (see Table 11 of [1]).**

**Answer**: Although the noiseless case does not yield significant improvement in the accuracy of clean data, the value of the proposed approach is in its significant robustness under noisy conditions and adversarial attacks, in addition to an uncertainty measure that reflects the model's self-confidence in its decision. That is, the value proposition of proposed approach is two-fold: 1) significant improvement in the accuracy under harsh noisy conditions and adversarial attacks, and 2) a learned uncertainty that reflects higher uncertainty - or decreased confidence - for erroneous predictions, which is not the case for other Bayesian and Ensemble approaches.

Employing a more advanced architecture would probably enhance the performance of all models. However, we ensured a fair comparison by employing the same architecture across all compared models. Our primary focus is not on achieving high accuracy, as we acknowledge that utilizing a more sophisticated architecture would likely yield improved performance. Rather, our primary focus is improving robustness, reliability, and self-awareness (through learned uncertainty quantification), under mild and harsher noise conditions and adversarial attacks.

**The authors provide an ablation over different numbers of particles. However, the number of models is still relatively high compared to standard ensemble methods.**

**Answer**: Selecting the number of particles in particle filtering approaches involves finding a balance between computational complexity and estimation accuracy. If the system exhibits high nonlinearity or multimodality, one might need a larger number of particles to accurately represent the posterior distribution.

First, we would like to clarify that our Basis-N for a given N is one model. N does not represent the number of models but represents the number of particles used to estimate the discrete predictive distribution of the model. That is, the output of our model is an entire distribution, rather than a point estimate, as in traditional ML methods.

Following the reviewer's suggestion, we included simulations considering a smaller number of particles, i.e., 5 and 10. In Fig. 8 of the revised manuscript, we compare all Basis-N models varying the number of particles N (N=5, 10, 100, 500, 1000) for Gaussian noise, targeted and untargeted Adversarial attacks. Interestingly, we observed that all N values resulted in equivalent performance for high SNR (SNR > 7). For low SNR, notice that the accuracy for Basis-5 is higher than that of the Ensemble obtained with 5 models; for example, for both targeted (SNR = 5) and untargeted (SNR =3) adversarial attacks, Basis-5 accuracy is higher than 75% (Fig. 8 (b) and (c)) while Ensemble accuracy is below 30% (Table 3).

**The y-labels in Figure 3 have an error.**

We thank the reviewer for noticing and prompting us to the error. We made sure that the y-labels are now correctly reported.

**In terms of the method itself, I have the following questions:**

**Eq. (9) introduces the gradient descent update rule as the transition model, which introduces a dependency of the transition model (Eq. 3) from W_k to W_{k+1} on the data y. Isn't this a problem for applying the SIS/PF model since the data is then essentially used twice - once in the transition- and once in the observation model?**

No, this does not present a problem because the transition model relies only on past data rather than current data. Below is a detailed explanation:

In Eq. (3), we outline the general state-space model. While $W_k$ appears as a function of $W_{k-1}$, it is important to note, by recurrence, that $W_{k-1}$ depends on all data up to time step (k-1). In our specific transition model described in Eq. (9), the gradient is taken with respect to $W_{k-1}$. In particular, the current time-step data is not used. We acknowledge that this detail was not sufficiently clear in the initial manuscript submission. The revised manuscript addresses this by correcting Eq. (9) for better clarity. Thank you for bringing this to our attention! Further, we have elaborated on Algorithm 1 and introduced notation for the gradient of the loss with respect to the previous time-step estimate, $W_{k-1}$.

The transition model in many PF approaches is chosen to be a random walk. However, for high dimensional state-space problems, the random walk may need exponentially high number of

particles or may not converge [9]. In general, the transition model is selected in a way to add information to aid the convergence. Some authors have proposed including likelihood terms in the importance distribution (transition model) to reduce the number of required particles and guide the few particles toward the regions of high likelihood [10, 11]. In the tracking literature, the use of gradient information in the transition model has been adopted in other PF approaches applied to various fields [12 - 14].

Furthermore, it is noteworthy that several authors in the literature have proposed the inclusion of "future/current" data in the transition density to direct particles towards areas influenced by future observations [11]. However, this approach would necessitate a re-derivation of the importance weights or ratios. We clarify that we are not employing this framework in our study. Nonetheless, we mention it for the reviewer's information.

[9] Snyder, Chris, et al. "Obstacles to high-dimensional particle filtering." Monthly Weather Review 136.12 (2008): 4629-4640.

[10] M. Ades, et al. "An exploration of the equivalent weights particle filter." Quarterly Journal of the Royal Meteorological Society 139.672 (2013): 820-840.

[11] Van Leeuwen, et al. "Nonlinear data assimilation in geosciences: an extremely efficient particle filter." Quarterly Journal of the Royal Meteorological Society 136.653 (2010): 1991-1999.

[12] A. Doucet, Arnaud, et al. "Parameter estimation in general state-space models using particle methods." Annals of the institute of Statistical Mathematics 55 (2003): 409-422.

[13] Tadic, Vladislav. "Asymptotic analysis of stochastic approximation algorithms under violated Kushner-Clark conditions with applications." Proceedings of the 39th IEEE Conference on Decision and Control (Cat. No. 00CH37187). Vol. 3. IEEE, 2000.

[14] V. B. Tadic. "Exponential forgetting and geometric ergodicity in state-space models." Proceedings of the 41st IEEE Conference on Decision and Control, 2002. Vol. 2. IEEE, 2002.

**On Page 7, the authors say that "to reduce computational complexity, [they] compute the gradient of the loss with respect to the previous time step weighted mean estimate rather than for each individual particle." This way of computing the gradient moves individual particles in a correlated manner. Does this break the underlying assumptions of the SIS/PF model?**

**Answer**: In the PF framework, there are no constraints on the choice of the transition model as long as it is Markovian. The importance density q, from which we sample, can also be anything as long as it has the same support as the posterior density [12, 15, 16]. In practice, both the transition and importance densities are strategically chosen to steer particles towards areas of high likelihood, thereby facilitating convergence. In our model, the transition density is the same as the importance density, and the traction term, distinguishing it from a mere random walk, is derived from gradient information. For instance, in reference [17], the authors demonstrate the application of Ensemble

Kalman Filters as the transition model, explicitly noting that this approach allows the current time-step particle i to be influenced by particles j from the previous time-step.

[15] D. Crisan and A. Doucet, "A survey of convergence results on particle filtering methods for practitioners," IEEE Trans. Signal Process., vol. 50, no. 3, pp. 736–746, Mar. 2002.

[16] M. S. Arulampalam, et al. "A tutorial on particle filters for online nonlinear/non-Gaussian Bayesian tracking." IEEE Transactions on signal processing 50.2 (2002): 174-188.

[17] P. J. Van Leeuwen, "Particle filters for the geosciences." Advanced Data Assimilation for Geosciences: Lecture Notes of the Les Houches School of Physics: Special Issue, June 2012 (2014): 291.

**Finally, I'd like to point out some sentences/claims that I suggest the authors to tone down a bit:**

**Page 4: "The network is not explicitly trained to learn uncertainty during the deterministic model training process." I would disagree with this statement since, e.g., [1] presents a method that combines ensemble methods with a VI loss that explicitly enforces a notion of diversity of model predictions on OOD data via a prior matching term in the loss.**

**Answer**: We deleted this sentence and replaced it by a more insightful discussion on prior networks. Please allow us to elaborate.

In the submitted paper, we wanted to highlight that classic deterministic training does not explicitly incorporate learning probability densities, even when a loss constraint is included to obtain confidence/uncertainty information. This means that, while the addition of a customized loss may modify certain aspects of the training, the fundamental mechanism remains unchanged, and the model is at most the couple (prediction, uncertainty). Notably, to the best of our knowledge, no work has derived a DL model that outputs a full predictive distribution.

In [18-19], Prior Networks do introduce the idea of training a network to get uncertainty estimates that can distinguish between in-distribution and out-of-distribution (OOD) data. However, this is done by explicitly training on not only the in-distribution data but also OOD data. As also pointed in [18], these OOD samples are usually unavailable and suggest that one should synthetically generate OOD data using a generative model or using an additional different dataset (e.g., FashionMNIST for MNIST). In contrast, we train a model to learn the entire predictive distribution based solely on the in-distribution data.

We understand that our sentence may have been misleading and added a discussion on prior networks. Please find the relevant portion in the revised version of the manuscript:

"Prior Networks (Malinin & Gales, 2018; 2019) offer an alternative to modeling uncertainty by allowing for data, distributional, and model uncertainty to be treated separately within a consistent, probabilistically interpretable framework. Prior Networks train a network to generate uncertainty

estimates that differentiate between in-distribution and out-of-distribution (OOD) data. This differentiation is achieved by training the network on both in-distribution and OOD data. However, as highlighted in (Malinin & Gales, 2018), OOD samples are often not readily available, leading to the suggestion that one should either synthetically generate OOD data using a generative model or employ an alternate dataset (e.g., using FashionMNIST (Xiao et al., 2017) for MNIST (Deng, 2012)) "

[18] A. Malinin, et al., "Predictive uncertainty estimation via prior networks," in Advances in Neural Information Processing Systems, 2018, pp. 7047–7058.

[19] Andrey Malinin, et al., "Reverse kl-divergence training of prior networks: Improved uncertainty and adversarial robustness," in Advances in Neural Information Processing Systems, 2019.

**Page 4: "Unlike the Variational Inference (VI) framework, our approach frees the posterior and predictive pdfs from any constraints on their form." I do not know if this claim can be made. Given that, technically speaking, SIS/PFs require a transition and observation model, the posterior (stationary) state distribution is restricted by precisely these assumptions.**

**Answer**: You rightly point out that the PF indeed operates within the structure provided by the transition and observation models, which can influence the form of the posterior distribution. Our aim was to emphasize that, unlike VI, where the form of the posterior is explicitly chosen from a specific family of distributions to simplify the computation, PF allows for a more flexible, empirical representation of the posterior through the use of particles. This, we believe, offers a form of adaptability in representing complex posterior distributions that might not easily fit into the predefined parametric forms typical of VI methods.

We acknowledge, however, that this adaptability is not without its own set of assumptions and constraints introduced by the choice of transition and observation models. To more accurately convey this nuance, we propose revising the statement to reflect the relative flexibility provided by our approach, while recognizing the inherent constraints posed by the necessary models. The revised version of the manuscript includes the following adjustment:

"Unlike the Variational Inference (VI) framework, which typically constrains the posterior and predictive pdfs to a predefined family of distributions, our approach employs a particle-based representation that offers a more flexible empirical approximation of these distributions. However, it is important to note that this flexibility is contextual and operates within the confines of the chosen transition and observation models."

**Requested Changes:**

**In terms of the experiments, I would like to see the following changes:**

**Addition of statistical uncertainty measures (e.g. standard deviation or -error, quantiles, ...) to the Figures and Tables. If the current data resulted from only one seed, I suggest running at least five seeds per dataset and method**.

For all experiments, we reported the 95% Confidence Intervals. We report the values in Tables 1 - 6 and included shaded regions for the plots in Figs. 1 - 5. As suggested by the reviewer, we included an analysis on different initializations and reported it in Table 9 in the Appendix. In addition, see detailed answer to Comment 1.

**Application of all attack schemes to all methods and datasets or an explanation of why a particular attack method does not apply to a specific model and dataset combination.**

Please see detailed answer to Comment 2. The revised version of the manuscript includes the following explanation in the revised manuscript:

"The study aims to showcase various attacks (FGSM and PGD targeted and untargeted) across diverse datasets (MNIST, CIFAR-10, Malaria, and Hippocampus), recognizing that attack effectiveness depends on the power of attack as well as inherent dataset characteristics. Experiments varied the attack intensity (measured by SNR) from high to low to comprehensively assess model resilience. We systematically explored a variety of attack scenarios by employing a mix of targeted and untargeted attacks across different datasets. This approach allowed us to showcase a broad range of adversarial situations."

**Ideally, an ablation over the sigma_{eta} parameter of the BaSIS-Net on one dataset. At least, a discussion of choosing the parameter and the observed sensitivity.**

Please see detailed answer to related comment 3. We added a discussion as suggested in section 3.4. We additionally performed an ablation study on MNIST and CIFAR-10 and reported it in the Appendix of the revised manuscript.

**CIFAR-10 experiments with a more standard network architecture (e.g., the one presented in [1]).**

Please see detailed answer to related comment 6.

**An ablation of BaSIS-Net with only 5, 10, and 20 parameters.**

Please see detailed answer to related comment 7.

We included the results for 5 an 10 particles.

**Comparison of uncertainty "calibratedness" (Figures 3, 4, 5, and 6) to baseline methods.**

Please see detailed answer to related comments 4 and 5.

**On a methodological level, I would like to see a discussion of the two questions on the instantiation of the SIS/PF framework. Such a discussion in Section 3.4 helps to avoid questions/misconceptions by readers**.

The revised version of the manuscript includes a thorough discussion and simulations on all questions/comments by this reviewers. In particular, the questions of:

**A revision of the two sentences I have highlighted above.**

We have revised the two sentences highlighted above. Please refer to detailed answers to comments 10 and 11.

**REVIEWER 5b5L**

Weaknesses:

**W1. By referencing the asymptotic convergence of Monte Carlo methods (among others at the end of Section 6.2), I think that in the current form the paper seems to be implying that the empirical success claimed in the Experiments section (in terms of classification performance under distribution shift) stems from the fact the inference method performs inference successfully. In my opinion, as it currently stands, with the current empirical evidence, that conclusion is not warranted. This is particularly true, e.g., due to some approximations are being made such as evaluating likelihood/loss not on per-particle basis but using a weighted mean estimate of the parameters at a given step in training, which seems to me like a significant modification of the SMC algorithm, and I'm not sure/I doubt the convergence guarantees apply in that case. I see two solutions: A) evaluate the quality of the approximate Bayesian inference achieved by evaluating against a benchmark such as [1], or B) make it explicit that the quality of Bayesian inference has not been assessed. [2] suggests that samples from a reasonably-converged Hamiltonian Monte Carlo chain, (supposedly) approximating the posterior better than other benchmarked approximate inference methods, exhibits lower distribution shift robustness than those other methods. Hence, I have doubts whether using the empirical success in the evaluation suite used can be used as an argument in support of the quality of the posterior approximation.**

**Answer**: Thank you for your insightful and detailed feedback and for highlighting the concerns regarding the implications drawn from the asymptotic convergence of Monte Carlo methods and the empirical success reported in our Experiments section. We acknowledge the points you've raised about the potential mismatch between the classification performance under noisy and adversarial conditions and the methodological adjustments we made, specifically, the use of the weighted mean estimate for evaluating likelihood/loss instead of a per-particle basis. We note that, in our PF algorithm, we do evaluate the likelihood per particle (step 7 in Algorithm 1). But we consider the weighted mean estimate when computing the gradient in the transition model (step 5 in Algorithm 1) primarily to reduce computational complexity. This approach allows us to compute a single gradient rather than N gradients for each particle.

Upon reflection, we agree that our current presentation may overstate the direct link between the empirical success of our model and the efficacy of the inference method, especially considering the small modifications we've implemented in the PF framework. We appreciate the reference to [2], which raises an important consideration regarding the relationship between posterior

approximation quality and distribution shift robustness. We added this reference to the revised version of the manuscript. To address your concern, we will consider option B) by revising the discussion in Section 6.2 to explicitly state that "while our empirical results demonstrate promising classification performance, the direct correlation to the quality of PF predictive inference, given the methodological modifications we've employed in evaluating likelihood/loss instead of a per-particle basis (step 5 in Algorithm 1), warrants further investigation [2]."

[2] Izmailov et al., What Are Bayesian Neural Network Posteriors Really Like?, 2021

**W2. The inference-time mechanism of BaSIS-Net is not well presented (unlike the training time algorithm), and I conclude I don't understand it. Authors say "BaSIS [...] unlike many post-hoc methods that necessitate multiple forward passes at inference time to calculate uncertainty" implying that at inference time BaSIS doesn't require multiple forward passes, but Figure 8 shows linear complexity with the number of particles. Isn't that somewhat contradictory?**

**Answer**: At inference time, sampling is not necessary because the particles and their ratios are readily available, allowing us to directly compute the full predictive distribution (see Fig. 7). Given this distribution, we can easily compute its associated moments, e.g., mean for point-estimate classification and the variance for the predictive uncertainty. This stands in contrast to other methods, which require sampling a set of parameters, passing them through the model to obtain a single prediction. Ensemble methods similarly repeat the forward pass but cycle through different models in the ensemble rather than sampling parameters.

We realize that the lack of a clear description of the inference mechanism may have led to confusion. To rectify this, we have introduced a new subsection, 3.4.1 'BaSIS Inference Mechanism,' within Section 3.3, to thoroughly explain the inference process, and ensure a better understanding of how BaSIS-Net efficiently handles inference.

**W3. Unlike the inference-time comparison (Table 7), the training-time comparison is not reported. I'd like to see it reported because I think it's as important to provide a full-picture of the method as inference-time comparison. I'd also like to see memory-cost comparison, because I think that's a criterium where BaSIS incurs a particularly large cost. I'd like to see all of these in a single section.**

**Answer**: Following the reviewer's suggestion, we added a new table (Table 7) in the updated version of the manuscript similar to that previously reported for the inference time (Table 8 in the updated manuscript). It is not surprising that our method is more computationally demanding as compared to other approaches. There is a trade-off between performance and computational complexity. Additionally, we discussed all the findings, including the memory-cost comparison in the Computational complexity section 6.4. Please find the relevant portion reported below:

"The computational demands of SIS and PFs algorithms can be substantial, particularly when the number of particles, N, increases. In particular, if the model has d parameters, PF requires storage of d N particles. For instance, in the MNIST architecture we used, a deterministic model and MC Drop would need 183.13 KB of storage using $4 \cdot d / 1024$ with $d = 46880$. Bayesian methods, such

as BCNN or VDP, would require roughly twice that amount because they have twice the number of parameters, i.e., means and variances. The storage requirement of Ensemble and Basis scales with the number of models and particles, respectively. This can lead to a trade-off between the number of particles and the performance of BaSIS.

The interplay between the performance of BaSIS model, the number of particles N, and computational demands is illustrated in Figs. 7 and 8, as well as Tables 7 and 8. The average inference time per image is plotted against the number of particles N of BaSIS-Net for MNIST in Fig. 8. It is observed that the test time approximately scales linearly with the number of particles N.

We present a comparison of the training time and inference time for all approaches in Tables 7 and 8, respectively.

Depending on the number of particles N and the specific dataset, the computational complexity of training BaSIS may be lower compared to VDP frameworks (exVI, enVI, unVI), which propagate a Taylor-series approximation of the mean and variance of the variational distribution. In general, other approaches, including MC Drop, BCNN, and Ensemble, are less computationally demanding. This difference arises from these models using point-estimate methods, where a single sample is propagated through the model layers. However, when employing these models as approximate Bayesian methods, multiple samples need to be considered at inference time to estimate uncertainty. For instance, the inference time of Basis with 10 particles is 0.10 ~$(10^{-3}$ s), whereas an ensemble of $10$ models would require $0.04 x·10 ~(10^{-3} s)$."

Training is conducted offline, which allows for flexibility in managing computational resources. Notably, Table 8 shows that the inference time for BaSIS with 100 particles is comparable to that of MC Dropout and ensemble methods involving approximately 10 to 20 models. This equivalency in inference time efficiency underscores an important trade-off: the computational complexity incurred during BaSIS's training phase is offset by several key advantages. Firstly, BaSIS demonstrates significant improved robustness in noisy and adversarial conditions, showcasing its resilience and reliability. Secondly, its design facilitates self-assessment capabilities through the utilization of the second moment of the predictive distribution, providing a deeper insight into the model's uncertainty. Lastly, BaSIS opens the door to exploring the impact and effects of higher-order moments on accuracy, robustness, and self-assessment. Although the exploration of higher-order moments falls beyond the scope of this paper, it highlights the promising potential of BaSIS for pioneering research in this domain. This upfront computational investment during training could be a worthwhile exchange for the gains achieved in model performance and analytical depth."

**W4. There exist more suitable metrics for investigating uncertainty quantification than accuracy (which doesn't capture it at all) - see Section 3 of [3]. For the sake of better characterization of the proposed method, could you computation of Brier score for all of the existing results, please?**

**Answer**: Thank you for suggesting the use of the Brier score. First, we would like to clarify that our primary metric for quantifying uncertainty is not based on accuracy. Our approach to

evaluating a robust measure of uncertainty is through distributional-shift analysis, where an effective uncertainty measure is expected to exhibit monotonic behavior in relation to the signal-to-noise (SNR) ratio at the input. Specifically, as the SNR deteriorates, the model's uncertainty in its predictions should correspondingly increase. We use accuracy in conjunction with uncertainty to demonstrate that the model exhibits low uncertainty (or high confidence) when accurate, and conversely, high uncertainty (or low confidence) when its predictions are incorrect. For instance, Figure 3 illustrates that as accuracy declines, the network concurrently produces higher variance values, indicating increased uncertainty with decreasing signal-to-noise ratios. We have added a comparative analysis with other methods in the revised manuscript to provide a comprehensive view.

Following established practices in the literature [3], our objective is to demonstrate that even when a model is "fooled" or makes erroneous predictions, it can still assign high uncertainty to those predictions, indicating an awareness of its own limitations [3]. We did not utilize the Brier score because our model provides a full predictive distribution, approximated by weighted particles, rather than a point estimate. This allows us to calculate all necessary point estimates, including predictions and variance, directly from the full distribution.

Following the reviewer's suggestion, we computed the Brier score. Specifically, Figure 5 shows the Brier score versus SNR for MNIST. We notice that when the accuracy decreases (for noise or attacks), the Brier score increases, suggesting that the model is less "to be trusted". Likewise, our uncertainty measure is revealing that in Fig. 3.

Our analysis operates under the premise that a valid uncertainty estimate should reflect lower confidence/higher uncertainty for images that are perturbed or subjected to attacks. Through our experiments, we explored how the calculated second moment (variance/uncertainty) responds to such perturbations, adding varying levels of Gaussian noise (SNR) to the test datasets and executing simulations with both targeted and untargeted adversarial attacks. The trustworthiness of the predictions is thus gauged by these uncertainty values.

The benefits of our approach are twofold: 1) it yields a significant enhancement in accuracy amidst noisy conditions and adversarial attacks; and 2) it produces a predictive variance, computed from a full predictive distribution, that accurately signals higher uncertainty or reduced confidence when facing incorrect decision-making or distributional shifts. This latter aspect distinguishes our method from other Bayesian approaches, which may not accurately reflect uncertainty under similar conditions.

**If you disagree with some of my assessments/comments, I hold my opinions not very strongly, and given evidence I am reasonably open to be convinced otherwise.**

Answer: Thank you for your openness and flexibility. We appreciate your willingness to consider alternative perspectives.

**Questions:**

**Q1. How is the second moment for the predictions in the classification context computed? The NN returns a vector of logits - what happens next?**

First, let us re-emphasize that, in the context of particle filtering, the output of the DL model is a comprehensive predictive distribution for every label or class. To elucidate the inference process in BaSIS, we note that in our trained BaSIS model, the optimal parameters of every layer $W_{[l]}^{*}$ are represented by N particles or samples, denoted as $\{W_{[l]}^{*(i)}\}\_{i=1}^N$, each assigned a uniform weight of 1/N due to the resampling step during training (see Eqs. (9-10) in Algorithm 1). Consequently, for every sample parameter $W_{[l]}^{*(i)}$, feature maps are generated at the output of every layer, resulting in N feature maps or "N feature map particles" (see Eq. (12)) (in contrast to a point-estimate feature map in deterministic models). Before moving to the next layer, we compute the mean estimate for these "feature map particles", which will serve as the input for the next layer. Specifically, for the "logit layer", there are N representations or particles for every logit vector, meaning the logit vector is essentially a matrix of size (c x N), where c is the number of classes (e.g., c = 10 for MNIST). Softmax is applied to each logit particle, producing a discrete distribution (represented by weighted N particles) for every label or class. This is how a full predictive distribution for every class is achieved at inference time (see Fig. 7). We have introduced Subsection 3.4.1, titled "BaSIS Inference Mechanism," to elucidate the inference process for BaSIS and explain how the model generates a full predictive distribution for every sample.

The model classification and uncertainty are computed as a post-inference step by using the predictive distribution. The uncertainty is obtained as the sample variance of this discrete predictive distribution.

Due to the double-blinded review process, we are unable to share our GitHub repository at this time. However, we are committed to sharing our code with the community as soon as the paper is accepted after review.

[1] Wilson et al., Evaluating Approximate Inference in Bayesian Deep Learning, 2022

[2] Izmailov et al., What Are Bayesian Neural Network Posteriors Really Like?, 2021

[3] Ovadia et al., Can You Trust Your Model's Uncertainty? Evaluating Predictive Uncertainty Under Dataset Shift, 2019

**Requested Changes:**

**Critical for securing my recommendation for acceptance: W1, W2, W3, W4, adding a baseline of the standard SGD-trained model on the tasks evaluated**.

**Nice to haves: SGLD baseline (according to [2] one of the strongest methods for distribution shift robustness)**

**Answer**: We included a standard model in all our evaluations (Tables 1 – 6 and Figs. 1 and 2). We refer to it as Det in the revised manuscript and in all tables and figures. Despite its similar performance (in terms of accuracy) to all other models under normal clean conditions, the standard deterministic model exhibits the poorest performance among all approaches under all noise and adversarial conditions. Unfortunately, given the time-constraint for the resubmission and the computational limitation, we could not include SGLD baseline for all experiments.

**Typos:**

We thank the reviewer for taking the time to point out the typos. We have addressed all of them in the revised version of the manuscript.

**REVIEWER jWLm**

We present our approach in section 3.4. However, to ensure the completeness and self-sufficiency of the paper and to facilitate a clear understanding of our approach, we provide detailed background concepts and derivations, e.g., particle filter and importance sampling in sections 3.1 – 3.3.

Please find below our replies to the reviewer's requested changes. We will update the revised version of the paper with these changes to improve the organization of Section 3.4.

*I'm confused about the algorithm. Specifically, I'm not sure how to combine the importance ratio from each layer.*

**Answer**: The problem of good approximation of the posterior probability distribution becomes increasingly challenging as the dimension of the state space increases; that is the number of particles necessary for tracking increases with the size of the space. To address this computational challenge, an effective approach is the concept of multiple particle filters as discussed in [1 In the context of DL models, we employed layer-wise partitioning to decompose the state-space of the model parameters into more manageable layer-wise sub-spaces.

It's important to note that the partitioning doesn't extend to the likelihood; instead, all observations are used to estimate each sub-space. The process involves initially estimating the parameters of the first layer under the assumption that all other layers are fixed at their previous time estimates. Subsequently, the second layer is estimated given the first layer at the current time and all other layers fixed at their previous time estimates, and so on until the last layer, considering all previously estimated parameters.

This approach implies that at each time instant, all particle filters receive predictions of their states and share this information with the remaining particle filters. The exchanged information is utilized to compute the weights of particles and generate new particles. Consequently, importance ratios are computed independently for each layer, eliminating the need for further combination of importance weights. In detail, given the layer-wise state calculations, the network's output is determined using the current sample particles for the considered layer, while all other layers are held constant at their most recent estimates.

If we let $l$ denote the layer, $L$ the total number of layers, $k$ the time-step, $i$ the particles, and $N$ the total number of particles, we define:

$$y_{[l]k}^{(i)} = f\big([W_k]^{(i)}, X\big) \text{ with } \quad [W_k]^{(i)} = \big\{W_{[1]k}, \ldots, W_{[l-1]k}, \ W_{[l]k}^{(i)}, W_{[l+1]k}, \ldots, W_{[L]k-1}\big\} i = 1, 2, \ldots, N$$

Consequently, the importance ratios can be computed separately for each layer as:

$$\tilde{u}_{[l]k}^{(i)} \propto u_{[l]k-1}^{(i)} e^{-\mathcal{L}(y_{[l]k}^{(i)})}$$

***I'm also confused about the motivation of using the loss to define the likelihood. For certain cases like cross-entropy loss and L2 loss, there are clear probabilistic interpretations. But for general cases, it is weird.***

**Answer**: In particle filters, the use of the loss to define the likelihood is grounded in Bayesian inference principles. The likelihood quantifies the probability of observed data given a specific state hypothesis, crucial for updating beliefs about the state based on new observations. By employing a loss function to measure the dissimilarity between predicted and observed values, the likelihood becomes a weighted measure of consistency, influencing the assignment of weights to particles. Higher likelihood, indicative of lower loss, results in higher particle weights, accentuating their impact on the posterior distribution estimate. This approach aligns with the adaptability of particle filters to diverse problem scenarios, where the choice of loss function can be tailored to the characteristics of the problem, ensuring flexibility in handling different types of observations and improving the accuracy of state estimation.

***I'm also confused about the gradient in the algorithm box. Specifically, what are the other layers' weights when take the gradient for certain layer.***

**Answer**: The gradient in the algorithm box adheres to the system dynamics outlined in Eq. (9):

$$\mathcal{W}_k = \mathcal{W}_{k-1} - \alpha \nabla_{\mathcal{W}} \mathcal{L},$$

where $\mathcal{W}_k = \{\mathcal{W}_{[1]k}, \mathcal{W}_{[2]k}, \dots, \mathcal{W}_{[L]k}\}$ is the partitioned state at time step $k$, $\nabla_{\mathcal{W}} \mathcal{L}$ is the gradient of the network loss function with respect to the parameters, and $\alpha$ is the learning rate.

In the context of layer $l$ at time $k$, $N$ particles are generated utilizing the $N$ particles from time $k-1$. The generation of each new particle involves propagating the previous particle in the direction opposite to the gradient of the loss. The objective is to acquire a new particle with a reduced loss, or equivalently, an increased likelihood. When computing the gradient for a specific layer, the weights of the other layers remain constant at their estimates from the previous time step. Consequently, only the parameters of the targeted layer undergo updates.

Following the reviewer's suggestion, we will change the notation in the algorithm:

Old version: $\mathcal{W}_{[l]k}^{(i)} = \mathcal{W}_{[l]k-1}^{(i)} - \alpha \nabla_{\mathcal{W}_{[l]k-1}} \mathcal{L} + \eta^{(i)}, \quad \eta^{(i)} \sim \mathcal{N}(0, \sigma_\eta^2), i = 1, 2, \dots, N$

New version: $\mathcal{W}_{[l]k}^{(i)} = \mathcal{W}_{[l]k-1}^{(i)} - \alpha \nabla_{\mathcal{W}_{[l]k-1}} \mathcal{L}(\mathcal{W}_{k-1}) + \eta^{(i)}, \quad \eta^{(i)} \sim \mathcal{N}(0, \sigma_\eta^2), i = 1, 2, \dots, N$

Where $\mathcal{W}_{k-1} = \{\mathcal{W}_{[1]k-1}, \mathcal{W}_{[2]k-1}, \dots, \mathcal{W}_{[L]k-1}\}$ is the state at time $k-1$.

**Requested Change 4**: ***I would like to ask for the specific form of the predictive distribution, and the consistency of it.***

**Answer**: Particle Filter (PF) represents a powerful class of sequential Monte Carlo algorithms that numerically solve the non-linear optimal estimation problem without making any assumptions about the form of the probability density functions or the linearity of the system model [2]. With PF, the posterior distribution is approximated by a discrete mass of weighted particles. The use of particles allows PF to handle complex, non-Gaussian, and nonlinear system dynamics, providing a flexible framework for state estimation in situations where traditional parametric methods may struggle. Additionally, PF converges under weak assumptions to the true distribution ($N \rightarrow \infty$), i.e., PF have been shown to converge almost surely toward the optimal filter as the number of particles increases [3].

[1] Djuric, Petar M., Ting Lu, and Mónica F. Bugallo. "Multiple particle filtering." 2007 IEEE International Conference on Acoustics, Speech and Signal Processing-ICASSP'07. Vol. 3. IEEE, 2007.

[2] Gordon, Neil J., David J. Salmond, and Adrian FM Smith. "Novel approach to nonlinear/non-Gaussian Bayesian state estimation." IEE proceedings F (radar and signal processing). Vol. 140. No. 2. IET Digital Library, 1993.

[3] Crisan, Dan, and Arnaud Doucet. "A survey of convergence results on particle filtering methods for practitioners." IEEE Transactions on signal processing

References: ABasic Convergence Result for Particle Filtering Xiao-Li Hu, Thomas B. Schön, Member, IEEE, and Lennart Ljung, Fellow, IEEE – use error between true and prediction for observation model 9for reviewer 3)

[1] FAST PARTICLE FILTERS FORMULTI-RATESENSORS Thomas B. Sch¨on, David T¨ornqvist and Fredrik Gustafsson – same error + show dependence of transition on input data.

[2] [12]Doucet, Arnaud, and Vladislav B. Tadić. "Parameter estimation in general state-space models using particle methods." Annals of the institute of Statistical Mathematics 55 (2003): 409-422.

[3] [13] Tadic, Vladislav. "Asymptotic analysis of stochastic approximation algorithms under violated Kushner-Clark conditions with applications." Proceedings of the 39th IEEE Conference on Decision and Control (Cat. No. 00CH37187). Vol. 3. IEEE, 2000.

[4] [14] TadiC, Vladislav B. "Exponential forgetting and geometric ergodicity in state-space models." Proceedings of the 41st IEEE Conference on Decision and Control, 2002.. Vol. 2. IEEE, 2002.

[5] [10]Ades An exploration of the equivalent weights particle filter. Quart. J. Royal

Meteorological Soc., 139(672):820–840, 2013

[6] [9]Snyder, Chris, et al. "Obstacles to high-dimensional particle filtering." Monthly Weather Review 136.12 (2008): 4629-4640.

[7] [11] Nonlinear data assimilation ingeosciences: anextremelyefficientparticlefilter.

[8] [17] Van Leeuwen, P. J. "Particle filters for the geosciences." Advanced Data Assimilation for Geosciences: Lecture Notes of the Les Houches School of Physics: Special Issue, June 2012 (2014): 291.

[9] [1]SENSITIVITY TO NOISE IN PARTICLE FILTERS FOR 2-D TRACKING ALGORITHMS dong

10 [2]Gordon, Neil J., David J. Salmond, and Adrian FM Smith. "Novel approach to nonlinear/non-Gaussian Bayesian state estimation." IEE proceedings F (radar and signal processing). Vol. 140. No. 2. IET Digital Library, 1993.

[11] [16] A Tutorial on Particle Filters for Online Nonlinear/Non-Gaussian Bayesian Tracking

[15] D. Crisan and A. Doucet, "A survey of convergence results on particle filtering methods for practitioners," IEEE Trans. Signal Process., vol. 50, no. 3, pp. 736–746, Mar. 2002.

[12] [3]Jungo, Alain, Fabian Balsiger, and Mauricio Reyes. "Analyzing the quality and challenges of uncertainty estimations for brain tumor segmentation." Frontiers in neuroscience (2020): 282.

[13] [4] Carannante, Giuseppina, et al. "Self-Assessment and Robust Anomaly Detection with Bayesian Deep Learning." 2022 25th International Conference on Information Fusion (FUSION). IEEE, 2022.

[15] [5]McCrindle, Brian, et al. "A Radiology-focused Review of Predictive Uncertainty for AI Interpretability in Computer-assisted Segmentation." Radiology: Artificial Intelligence 3.6 (2021).

[16] [18]Andrey Malinin and Mark Gales, "Predictive uncertainty estimation via prior networks," in Advances in Neural Information Processing Systems, 2018, pp. 7047–7058.

[17] [19]Andrey Malinin and Mark JF Gales, "Reverse kl-divergence training of prior networks: Improved uncertainty and adversarial robustness," in Advances in Neural Information Processing Systems, 2019.