# OpenReview forum: "BaSIS-Net: From Point Estimate to Predictive Distribution in Neural Networks - A Bayesian Sequential Importance Sampling Framework"
_TMLR — Accepted by TMLR_

### Review · Reviewer_jWLm · 2023-12-26

**Summary Of Contributions:**

This manuscript proposed a method for Bayesian neural networks using a sequential importance sampling based method.

**Audience:**

Yes

**Broader Impact Concerns:**

No.

**Claims And Evidence:**

No

**Requested Changes:**

* I'm confused about the algorithm. Specifically, I’m not sure how to combine the importance ratio from each layer.
* I’m also confused about the motivation of using the loss to define the likelihood. For certain cases like cross-entropy loss and $L_2$ loss, there are clear probabilistic interpretations. But for general cases, it is weird.
* I’m also confused about the gradient in the algorithm box. Specifically, what are the other layers’ weights when take the gradient for certain layer.
* I would like to ask for the specific form of the predictive distribution, and the consistency of it.

**Strengths And Weaknesses:**

* Weaknesses
* * Section 3.1-3.3 are known to the community, and I believe the main novelty is at Section 3.4. However, Section 3.4 is not well-organized.

---

> ### Author Response · Authors · 2024-03-25
> **We addressed each comment and requested change point by point**
>
> We present our approach in section 3.4. However, to ensure the completeness and self-sufficiency of the paper and to facilitate a clear understanding of our approach, we provide detailed background concepts and derivations, e.g., PF and importance sampling in sections 3.1 – 3.3.
> 1 The problem of good approximation of the posterior probability distribution becomes increasingly challenging as the dimension of the state space increases; that is the number of particles necessary for tracking increases with the size of the space. To address this computational challenge, an effective approach is the concept of multiple particle filters as discussed in  Djuric et al. (2007). In the context of DL models, we employed layer-wise partitioning to decompose the state-space of the model parameters into more manageable layer-wise sub-spaces.
> The partitioning doesn't extend to the likelihood; instead, all observations are used to estimate each sub-space. The process involves initially estimating the parameters of the first layer under the assumption that all other layers are fixed at their previous time estimates. Subsequently, the second layer is estimated given the first layer at the current time and all other layers fixed at their previous time estimates, and so on until the last layer, considering all previously estimated parameters.
> This approach implies that at each time instant, all PF receive predictions of their states and share this information with the remaining PF. The exchanged information is utilized to compute the weights of particles and generate new particles. Consequently, importance ratios are computed independently for each layer, eliminating the need for further combination of importance weights. In detail, given the layer-wise state calculations, the network's output is determined using the current sample particles for the considered layer, while all other layers are held constant at their most recent estimates. The importance ratios can be computed separately for each layer as in step 8 Alg. 1
> 2 In PF, the use of the loss to define the likelihood is grounded in Bayesian inference principles. The likelihood quantifies the probability of observed data given a specific state hypothesis, crucial for updating beliefs about the state based on new observations. By employing a loss function to measure the dissimilarity between predicted and observed values, the likelihood becomes a weighted measure of consistency, influencing the assignment of weights to particles. Higher likelihood, indicative of lower loss, results in higher particle weights, accentuating their impact on the posterior distribution estimate. This approach aligns with the adaptability of particle filters to diverse problem scenarios, where the choice of loss function can be tailored to the characteristics of the problem, ensuring flexibility in handling different types of observations and improving the accuracy of state estimation.
> 3 The gradient in the algorithm box adheres to the system dynamics outlined in Eq.9
> In the context of layer l at time k, N particles are generated utilizing the N particles from time k-1. The generation of each new particle involves propagating the previous particle in the direction opposite to the gradient of the loss. The objective is to acquire a new particle with a reduced loss (increased likelihood). When computing the gradient for a specific layer l, the weights of the other layers remain constant at their estimates from the previous time step. Consequently, only the parameters of the targeted layer undergo updates.
> The gradient is taken with respect to $W_{[l]k-1}$. We acknowledge that this detail was not sufficiently clear in the initial manuscript submission. The revised manuscript addresses this by correcting step 5 in Alg. 1.
> 4 PF represents a powerful class of sequential MC algorithms that numerically solve the non-linear optimal estimation problem without making any assumptions about the form of the pdf or the linearity of the system (Gordon et al., 1993; Doucet et al., 2001). With PF, the posterior pdf is approximated by a discrete mass of weighted particles. The use of particles allows PF to handle complex, non-Gaussian, and nonlinear system dynamics, providing a flexible framework for state estimation in situations where traditional parametric methods may struggle. Additionally, PF converges under weak assumptions to the true distribution (N→∞), i.e., PF have been shown to converge almost surely toward the optimal filter as the number of particles increases (Crisan & Doucet, 2002).

---

### Review · Reviewer_5b5L · 2024-01-18

**Summary Of Contributions:**

The authors propose a new method for uncertainty quantification for neural networks (NN) based on Sequential Monte Carlo (SMC) named BaSIS-Net.
The “final” posterior over the model’s weights is approximated by a set of weighted particles, as typical for SMC, in contrast to unweighted samples (e.g. using some form of MCMC inference) or a parametric distribution when using variational inference.
The problem of posterior inference in NN is casted as a problem of inference in a state-space model - the posterior over the state at time $t$ corresponds to the posterior over the weights after $t$ gradient updates of the NN - and the particles approximating the posterior are updated and resampled at each gradient step over the course of training.
On the set of evaluations the authors chosen, the method they propose yields competitive performance wrt several baseline methods from the field of Bayesian Deep Learning literature.

**Audience:**

Yes

**Claims And Evidence:**

No

**Requested Changes:**

Critical for securing my recommendation for acceptance: W1, W2, W3, W4, adding a baseline of the standard SGD-trained model on the tasks evaluated.

Nice to haves: SGLD baseline (according to [2] one of the strongest methods for distribution shift robustness)

Typos:
* page 3: “density estimation techniques NN training”, “estimates f the gradient”
* Figure 8: instead of $s^-3$ maybe $10^-3 s$ or simply $ms$?

**Strengths And Weaknesses:**

Strengths:
S1. have never seen SMC applied to Bayesian Deep Learning (BDL) inference, and so I consider it novel. What is more, if someone asked me apriori whether this approach is going to work, I’d seriously doubt the inference done this way would succeed in reasonably approximating to the posterior (and I still have those doubts as that is not directly evaluated by the authors, see W1), so it is interesting to see that it results in competitive performance on a set of tasks the authors evaluate on (classification performance under distribution shift).
S2. The paper is mostly well written and easy to read, however I am missing several important pieces of information, see W2 & W3.

Weaknesses:
W1. By referencing the asymptotic convergence of Monte Carlo methods (among others at the end of Section 6.2), I think that in the current form the paper seems to be implying that the empirical success claimed in the Experiments section (in terms of classification performance under distribution shift) stems from the fact the inference method performs inference successfully. In my opinion, as it currently stands, with the current empirical evidence, that conclusion is not warranted. This is particularly true, e.g., due to some approximations are being made such as evaluating likelihood/loss not on per-particle basis but using a weighted mean estimate of the parameters at a given step in training, which seems to me like a significant modification of the SMC algorithm, and I’m not sure/I doubt the convergence guarantees apply in that case. I see two solutions: A) evaluate the quality of the approximate Bayesian inference achieved by evaluating against a benchmark such as [1], or B) make it explicit that the quality of Bayesian inference has not been assessed. [2]  suggests that samples from a reasonably-converged Hamiltonian Monte Carlo chain, (supposedly) approximating the posterior better than other benchmarked approximate inference methods, exhibits lower distribution shift robustness than those other methods. Hence, I have doubts whether using the empirical success in the evaluation suite used can be used as an argument in support of the quality of the posterior approximation.
W2. The inference-time mechanism of BaSIS-Net is not well presented (unlike the training time algorithm), and I conclude I don’t understand it. Authors say “BaSIS [...] unlike many post-hoc  methods that necessitate multiple forward passes at inference time to calculate uncertainty” implying that at inference time BaSIS doesn’t require multiple forward passes, but Figure 8 shows linear complexity with the number of particles. Isn’t that somewhat contradictory?
W3. Unlike the inference-time comparison (Table 7), the training-time comparison is not reported. I’d like to see it reported because I think it’s as important to provide a full-picture of the method as inference-time comparison. I’d also like to see memory-cost comparison, because I think that’s a criterium where BaSIS incurs a particularly large cost. I’d like to see all of these in a single section.
W4. There exist more suitable metrics for investigating uncertainty quantification than accuracy (which doesn’t capture it at all) - see Section 3 of [3]. For the sake of better characterization of the proposed method, could you computation of Brier score for all of the existing results, please?

If you disagree with some of my assessments/comments, I hold my opinions not very strongly, and given evidence I am reasonably open to be convinced otherwise.

Questions:
Q1. How is the second moment for the predictions in the classification context computed? The NN returns a vector of logits - what happens next?

[1] Wilson et al., Evaluating Approximate Inference in Bayesian Deep Learning, 2022
[2] Izmailov et al., What Are Bayesian Neural Network Posteriors Really Like?, 2021
[3] Ovadia et al., Can You Trust Your Model’s Uncertainty? Evaluating Predictive Uncertainty Under Dataset Shift, 2019

---

> ### Author Response · Authors · 2024-03-25
> **We addressed all comments point by point**
>
> 1 We acknowledge the points you've raised about the potential mismatch between the classification performance under input shifts and the methodological adjustments we made: use of the weighted mean estimate for evaluating likelihood/loss instead of a per-particle basis. In our PF algorithm, we evaluate the likelihood per particle (step 7 Alg. 1). But we consider the weighted mean estimate for the gradient in the transition model (step 5 Alg. 1) primarily to reduce computational complexity. This approach allows us to compute a single gradient rather than N gradients for each particle. We agree that our current presentation may overstate the direct link between the empirical success of our model and the efficacy of the inference method, especially considering the small modifications we've implemented in the PF framework. We revised Section 6.2.
>
> 2 At inference time, sampling is not necessary because the particles and their ratios are readily available, allowing us to directly compute the full predictive distribution. Given this distribution, we can easily compute its associated moments, mean for point-estimate classification and the variance for the predictive uncertainty.  We realize that the lack of a clear description of the inference mechanism may have led to confusion. To rectify this, we added 3.4.1 BaSIS Inference Mechanism.
>
> 3 We added Table 7 for training time. It is not surprising that our method is more computationally demanding as compared to other approaches. There is a trade-off between performance and computational complexity. We discuss all the findings, including the memory-cost comparison in 6.4.
>
> 4 Our primary metric for quantifying uncertainty is not based on accuracy. We operate under the premise that valid uncertainty estimates should reflect lower confidence/higher uncertainty for perturbed or attacked inputs. Through our experiments, we explored how the calculated second moment (variance/uncertainty) responds to such perturbations, adding varying levels of Gaussian noise (SNR) to the test datasets and executing simulations with both targeted and untargeted adversarial attacks. The trustworthiness of the predictions is thus gauged by these uncertainty values.  An effective uncertainty measure is expected to exhibit monotonic behavior in relation to the SNR levels. Specifically, as the SNR decreases, the model's uncertainty in its predictions should correspondingly increase. We added a comparative analysis with other methods in the revised manuscript. Our objective is to demonstrate that even when a model is “fooled” or makes erroneous predictions, it can still assign high uncertainty to those predictions, indicating an awareness of its own limitations [3]. We did not utilize the Brier score because our model provides a full predictive distribution, approximated by weighted particles, rather than a point estimate. Following the reviewer’s suggestion, we computed the Brier score. Figure 5 shows the Brier score versus SNR for MNIST. We notice that when the accuracy decreases (for noise or attacks), the Brier score increases, suggesting that the model is less “to be trusted”. Likewise, our uncertainty measure is revealing that in Fig. 3.
>
> Q1. In the context of PF, the output of the DL model is a comprehensive predictive distribution for every label or class. To elucidate the inference process in BaSIS, we note that in our trained BaSIS model,  the optimal parameters of every layer $W_{[l]}$ are represented by N particles, denoted as $W^i_{[l]}$ i =1, ...N, each assigned a uniform weight of 1/N due to the resampling step during training. Consequently, for every sample parameter $W^i_{[l]}$, feature maps are generated at the output of every layer, resulting in N feature maps or “N feature map particles” (see Eq. (12)) (in contrast to a point-estimate feature map in deterministic models). Before moving to the next layer, we compute the mean estimate for these “feature map particles”, which will serve as the input for the next layer. Specifically, for the “logit layer”, there are N representations or particles for every logit vector, meaning the logit vector is essentially a matrix of size (c x N), where c is the number of classes. Softmax is applied to each logit particle, producing a discrete distribution (represented by weighted N particles) for every label or class. This is how a full predictive distribution for every class is achieved at inference time. We have introduced Subsection 3.4.1, titled “BaSIS Inference Mechanism,” to elucidate the inference process for BaSIS and explain how the model generates a full predictive distribution for every sample.
> The model classification and uncertainty are computed as a post-inference step by using the predictive distribution.  The uncertainty is obtained as the sample variance of this discrete predictive distribution.
>
> We included a standard SGD in our evaluations. Given the time-constraint for resubmission, we couldn't include SGLD.

---

> > ### Comment · Reviewer_5b5L · 2024-04-03
> >
> > Thank you for your response.
> >
> > W1. Thank you for the updates.
> >
> > W2. Thank you for the updates.
> >
> > > Authors say “BaSIS [...] unlike many post-hoc methods that necessitate multiple forward passes at inference time to calculate uncertainty”
> >
> > Given section 3.4.1, it is now clear to me that you do perform multiple forward passes at inference time - you need to compute the forward pass for each of N particles in each layer, as per eq (12). Please remove this statement, I stand by the fact it's factually incorrect.
> >
> > On a different note, it seems odd to me that you take a mean across particles after each layer.
> > Wouldn't we expect to obtain a more accurate approximation of the predictive distribution if instead, we were to pass the N variants of the intermediate features (for each particle of the first layer's weights) independently through the N particles of the following layers, without performing the mean-reduction after each layer?
> > Especially since such procedure doesn't seem to incur a significantly larger computational cost.
> >
> > W3. Absolutely agree - I do not find it surprising either. Thank you for the updates.
> >
> > W4. Thank you for the updates.
> >
> > Q1. Following your description: assume `logits` is the "matrix of size (c x N)" you described.
> > Then `probs = logsoftmax(logits, dim=0)` to perform the softmax over the "particle" dimension.
> > When you say "the sample variance of this discrete predictive distribution" you mean something like `np.var(probs[y_true_label, :])`?

---

> > > ### Author Response · Authors · 2024-04-21
> > > **Responses point-by-point**
> > >
> > > 1) We apologize for any confusion caused by the statement in question and thank you for bringing it to our attention. We wanted to highlight that our inference procedure does not require the “classical” multiple forward passes where we sample a set of parameters each time. We note that, in our BaSIS model, the optimal parameters in every layer $l$, $W_{[l]}^{*}$, are represented by $N$ particles or samples (the substate vectors are represented by the layers). But to avoid any confusion and following the reviewer's suggestion, we removed the highlighted sentence.
> > >
> > >
> > > 2) We would potentially obtain a more accurate approximation of the predictive distribution by passing the $N$ variants of the intermediate features (for each particle of the first layer's weights) independently through the $N$ particles of the following layers. However, this process will incur a significantly larger computational cost that scales as $N$ to the power $L$, where $L$ is the number of layers. As an example, let us consider a three-layer model. Suppose we obtained $N$ feature map particles output of layer 1, $A_{[1]}^{(i)}$, $i = 1, …, N$, and suppose now we need to perform convolution with the $N$ particles of layer 2.  Each resulting feature maps, let’s say $A_{[2]}^{(i)}$, represents the product of a sample from the first distribution with a sample from the second distribution, resulting in $N$x$N$ feature maps particles that will need to be propagated to a third layer. Hence, at the third layer we would have $N$x$N$x$N$. This expansion can lead to a significantly larger computational cost, especially in deeper networks with many layers.
> > >
> > >
> > > 3) Yes, your provided pseudo code is overall correct. Let us clarify. At the “logit layer”, there are $N$ representations or particles for every logit vector, matrix of size ($c$ x $N$), where $c$ is the number of classes (e.g., $c = 10$ for MNIST). Softmax is applied to each logit particle as you would do normally (i.e., 0 if the matrix is $c$ x $N$).
> > > The uncertainty is obtained as the sample variance of this discrete predictive distribution. In particular, we consider v = np.var(probs, axis = -1). However, we then analyze the predictive variance intended as the variance value associated with the predicted label not the ground-truth label, i.e., predict_var = v[predict], with predict obtained from the mean prediction, predict = np.argmax(np.mean(probs, axis =-1))

---

### Review · Reviewer_X46h · 2024-02-24

**Summary Of Contributions:**

The paper presents an ensemble approach to Bayesian neural networks motivated by a particle filtering perspective. The presented approach is conceptually straightforward and nonetheless shows convincing performance in various experiments, demonstrating calibrated uncertainty estimation and solid performance.

**Audience:**

Yes

**Claims And Evidence:**

Yes

**Requested Changes:**

* In terms of the experiments, I would like to see the following changes:
  * Addition of statistical uncertainty measures (e.g. standard deviation or -error, quantiles, ...) to the Figures and Tables. If the current data resulted from only one seed, I suggest running at least five seeds per dataset and method.
  * Application of all attack schemes to all methods and datasets or an explanation of why a particular attack method does not apply to a specific model and dataset combination.
  * Ideally, an ablation over the sigma_{eta} parameter of the BaSIS-Net on one dataset. At least, a discussion of choosing the parameter and the observed sensitivity.
  * CIFAR-10 experiments with a more standard network architecture (e.g., the one presented in [1]).
  * An ablation of BaSIS-Net with only 5, 10, and 20 parameters.
  * Comparison of uncertainty "calibratedness" (Figures 3, 4, 5, and 6) to baseline methods.
* On a methodological level, I would like to see a discussion of the two questions on the instantiation of the SIS/PF framework. Such a discussion in Section 3.4 helps to avoid questions/misconceptions by readers.
* A revision of the two sentences I have highlighted above.

With these changes made, the paper is a solid contribution to the literature on approximate Bayesian neural networks that can motivate future research in applying the SIS/PF framework to this problem setting.

**Strengths And Weaknesses:**

Overall, the paper is well written, and the Sequential Importance Sampling (SIS) / Particle Filter (PF) discussion presents a sufficient overview for the rest of the paper. In general, I appreciate the simplicity of the approach and the primarily clear motivation from a SIS/PF perspective. The results in the experimental section demonstrate that the proposed approach leads to comparable or better results in robustness against (adversarial) noise. Finally, Figures 3 and 4 show a promising calibrated behavior of predictive uncertainty.

Having highlighted the paper's strengths, I want to list aspects in which the paper can improve. Starting with the experimental section, I see the following issues:
  * The figures and tables in the paper lack uncertainty intervals and, furthermore, information about how many seeds the authors averaged in the results.
  * The authors do not motivate why they do not attack all models with all schemes (i.e. (un-)targeted FSGM and PGD attacks) in Section 5.2
  * BaSIS-Net sed different noise values sigma_{eta} for different datasets and network architectures. However, the authors do not discuss the sensitivity of the method to this hyperparameter or how to choose it
  * Although the authors present evidence for the calibrated uncertainty of their method, they do not compare the "calibratedness" with the baseline approaches.
  * While interesting, Figures 5 and 6 are rather qualitative and lack a quantitative aspect that enables a comparison with baseline methods.
  * The authors use a custom architecture for the CIFAR-10 task instead of more standard architectures like ResNet-18, leading to significantly worse performance on noise-free images than already shown in the context of Bayesian NNs (see Table 11 of [1]).
  * The authors provide an ablation over different numbers of particles. However, the number of models is still relatively high compared to standard ensemble methods.
  * The y-labels in Figure 3 have an error.

In terms of the method itself, I have the following questions:
  * Eq. (9) introduces the gradient descent update rule as the transition model, which introduces a dependency of the transition model (Eq. 3) from W_k to W_{k+1} on the data y. Isn't this a problem for applying the SIS/PF model since the data is then essentially used twice - once in the transition- and once in the observation model?
  * On Page 7, the authors say that "to reduce computational complexity, [they] compute the gradient of the loss with respect to the previous time step weighted mean estimate rather than for each individual particle." This way of computing the gradient moves individual particles in a correlated manner. Does this break the underlying assumptions of the SIS/PF model?

Finally, I'd like to point out some sentences/claims that I suggest the authors to tone down a bit:
  * Page 4: "The network is not explicitly trained to learn uncertainty during the deterministic model training process." I would disagree with this statement since, e.g., [1] presents a method that combines ensemble methods with a VI loss that explicitly enforces a notion of diversity of model predictions on OOD data via a prior matching term in the loss.
  * Page 4: "Unlike the Variational Inference (VI) framework, our approach frees the posterior and predictive pdfs from any constraints on their form." I do not know if this claim can be made. Given that, technically speaking, SIS/PFs require a transition and observation model, the posterior (stationary) state distribution is restricted by precisely these assumptions.

---

> ### Author Response · Authors · 2024-03-25
> **We addressed each comment point by point and addressed the requested changes**
>
> 1PF isn't a traditional ML ensemble method but a technique to learn or track the posterior distribution of parameters given data. BaSIS-Net's output is the entire predictive distribution (as in Eq. 1) rather than a point-estimate. The reviewer's concern about using seeds to confirm model reliability is acknowledged. All experiments in this paper utilized random seeds. To address the concern, we trained 10 BaSIS-Net models, each with a different seed, with results reported in the appendix. We've also included 95% confidence intervals for all experiments in the revised manuscript.
>
> 2 We added a more detailed discussion in section 4.3 page 4. Our aim is to cover a wide array of attacks across diverse datasets. The attack effectiveness varies based on dataset characteristics. Experimentally, we employed attacks spanning different SNR to evaluate models' robustness. To maintain brevity and avoid an overly lengthy paper, we cannot display every dataset under every attack model.
>
> 3In state-space models, $\sigma_{\eta}$ governs the search space for particles at the next time step. Section 3.4.2 and the Appendix contain discussion and results related to the role of $\sigma_{\eta}$.
>
> 4The revised manuscript compares our method with others in Figs. 3,4,6. An effective uncertainty measure should convey higher uncertainty for low-accuracy predictions and capture shifts in input data.
>
> 5 Model uncertainty is captured by the variance of the predictive pdf. For segmentation, a predictive pdf is generated for each pixel. The variance of each pixel creates a pixel-level "uncertainty map" . We visualize this using a heatmap to avoid displaying a large matrix.
> Fig. 6 (revised manuscript) demonstrates a representative case from the Hippocampus dataset. This figure includes now a comparison with other methods.
> Our analysis explores input perturbations' effect on BaSIS-Net's predictive pdf. Fig. 7 is indeed qualitative and illustrates changes in the pdf when perturbations are applied to inputs. We plot the number of particles on the y-axis against the softmax scores on the x-axis.
>
> 6 The value of BaSIS is: significant improvement in the accuracy under perturbations, and a learned calibrated uncertainty.
> While a more advanced architecture could enhance all models performance, we ensured fair comparison by using the same architecture. Our focus isn't solely on achieving high accuracy; rather, it's on enhancing robustness, reliability, and self-awareness under various conditions.
>
> 7 Selecting the number of particles in particle filtering approaches involves finding a balance between computational complexity and estimation accuracy. If the system exhibits high nonlinearity or multimodality, one might need a larger number of particles to accurately represent the posterior distribution.
> We included simulations considering a smaller number of particles, i.e., 5 and 10 (Fig. 8).
>
> 8 y-labels are now correct.
>
> 9 No, this isn't an issue as the transition model relies solely on past data rather than current data. In our specific transition model (Eq. 9), the gradient is taken with respect to $W_{k-1}$, not the current time-step data. We've realized that this detail wasn't clear initially, so the revised manuscript corrects Eq. 9 for improved clarity.
>
> 10 In the PF framework, there are no constraints on the choice of the transition model as long as it is Markovian. The importance density, from which we sample, can also be anything as long as it has the same support as the posterior density (Crisan & Doucet, 2002). In practice, both the transition and importance densities are strategically chosen to steer particles towards areas of high likelihood, thereby facilitating convergence. In our model, the transition density is the same as the importance density, and the traction term, distinguishing it from a mere random walk, is derived from gradient information.
>
> 11 We deleted this sentence and replaced it by a more insightful discussion on prior networks in 2.1.
> We wanted to highlight that classic deterministic training does not explicitly incorporate learning probability densities, even when a loss constraint is included to obtain confidence/uncertainty information, and the model output is at most the couple (prediction, uncertainty). We understand that our sentence may have been misleading and added a discussion on prior networks in 2.1.
>
> 12PF indeed operates within the structure provided by the transition and observation models, which can influence the form of the posterior distribution. To more accurately convey this nuance, we propose revising the statement (end page 4).However, unlike VI, which explicitly chooses the posterior form from a specific family of distributions for computational simplicity, PF allows for a more flexible, empirical representation through particles. This adaptability offers a way to represent complex posterior distributions that may not easily fit predefined parametric forms typical of VI methods.

---

### Decision · Action_Editor_yhr3 · 2024-05-20

**Recommendation:** Accept with minor revision

**Comment:**

The author made a detailed response as well as updated significantly the manuscript to address most points from the reviewers. After their revision, reviewer 5b5L thought that the problematic claims had been addressed and recommended acceptance of the updated manuscript. On the other hand, both reviewers X46h and jWLm recommended rejection. Reviewer jWLm did not provide much explanation to justify to reject. Reviewer X46h still had unaddressed concerns about the manuscript; in particular, they stated they wanted all attacks on all datasets be included in the paper, as well as a more SOTA architecture employed on the CIFAR dataset. While these changes would improve the significance of the paper, I don't think they are required to ensure that the claims made by the authors are sufficiently supported, and thus I still recommend acceptance. The authors are encouraged to include these changes in the minor revision of the paper if they want to strengthen it, though it is not required.

What is required for the minor revision is to address the following other point made by reviewer X46h in their recommendation:
> The new tables 10 and 11 in the appendix provide valuable insights into the hyperparameter of BasisNet. However, the provided data raises a substantial new question: The model performance continuously increases with a decrease in $\sigma_n$. For the limit $\sigma_n=0$, the gradient computation via the particle mean would, however, raise questions about the importance of the PF framework for the performance of BasisNet. Interestingly, the authors did not choose the smallest investigated for the MNIST dataset. Is this because of a worse uncertainty calibration for lower values of $\sigma_n$? A thorough investigation of uncertainty calibration in dependence of $\sigma_n$ seems necessary to clarify these questions.

**Audience:**

Yes.

**Claims And Evidence:**

Yes.

---

> ### Author Response · Authors · 2024-06-10
> **Detailed Answer:**
>
> We appreciate the insightful comments regarding the new tables 10 and 11 in the appendix and the observations about the model performance in relation to the hyperparameter $\sigma_{\eta}$. Your point addresses the crucial trade-off between accuracy in clean conditions, robustness and uncertainty calibration, as illustrated in the new simulations shown in Fig. 11. Please allow us to elaborate below.
>
> Indeed, as $\sigma_{\eta}$ decreases, the test accuracy increases, but this comes at the expense of robustness and uncertainty calibration, as illustrated in the new Fig. 11. To clarify, when $\sigma_{\eta}$ becomes smaller, particle diversity is reduced, effectively making all particles equivalent and rendering the approach similar to a deterministic model. Our experiments (referenced throughout the paper and the new Fig. 11) demonstrate that the strength of our work lies not in achieving higher test accuracy under clean conditions—where deterministic models often perform comparably or better - but in enhancing robustness under noisy conditions and providing valuable uncertainty information through the propagation of particles in the model.
>
> In general, the choice of $\sigma_{\eta}$ is problem-specific and depends on the data, the model, and the desired balance between test accuracy under clean conditions and increased robustness with calibrated uncertainty. To validate this hypothesis, we studied the impact of $\sigma_{\eta}$ on accuracy under noisy conditions and on predictive uncertainty. Figure 11 shows the impact of transition noise $\sigma_{\eta}$ on test accuracy and predictive variance in MNIST test data. The different $\sigma_{\eta}$ selections are represented by the various colors. For the normalized predictive variance, solid lines represent correct classifications, and dashed lines represent incorrect ones. Note that: i) the smaller the value of $\sigma_{\eta}$, the smaller the gap between the predictive variances of correct and incorrect classifications. For the lowest $\sigma_{\eta}$  (shown in green), there is no difference in variance between correct and incorrect classification. This indicates that the calibration goodness of the uncertainty decreases with decreasing values of $\sigma_{\eta}$.
>
> Please find below the relevant excerpt from the modified discussion on this topic in the revised manuscript (Only the text in bold is new):
>
> “ … When starting our experiments, for each dataset, we performed a grid-search over a set of values for $\sigma_{\eta}$, starting from large to small, to identify a value that guarantees convergence with a good test accuracy. Notably, at comparable clean test accuracy, it is preferable to select a larger $\sigma_{\eta}$. When selecting small $\sigma_{\eta}$ values, the variation among particles decreases, leading to a model that behaves almost deterministically. In Appendix A, we report the accuracy of BaSIS-Net versus $\sigma_{\eta}$ on the MNIST and CIFAR-10 dataset as a proof of concept.
> **Additionally, we investigated the robustness and uncertainty calibration of the model with varying $\sigma_{\eta}$. Figure 11 shows test accuracy and predictive variance under Gaussian noise applied to MNIST test data, with several BaSIS-Net models using different $\sigma_{\eta}$ values.**”
>
> In the Appendix of the revised manuscript, we will also include the following discussion and results:
>
> **“We evaluated the impact of the transition noise parameter $\sigma_{\eta}$ on the robustness of BaSIS-Net to noisy conditions and assessed the predictive uncertainty of the resulting models. Our findings indicate that smaller $\sigma_{\eta}$ values are associated with higher accuracy on clean noiseless data. However, at higher noise levels, models with larger $\sigma_{\eta}$ exhibit more robust performance. Additionally, for low $\sigma_{\eta}$ values, the uncertainty estimates are not as informative as in models trained with larger values.**
>
> **In Fig. 11, we illustrate how test accuracy and variance information vary with different $\sigma_{\eta}$ selections under noisy conditions applied to the MNIST test data. We observe that models trained with a small $\sigma_{\eta}$ are less robust at low SNR, and their variance is unable to distinguish between correctly and incorrectly classified inputs. This undermines one of the key features of our BaSIS-Net framework, which is the ability to provide meaningful uncertainty estimates that differentiate between correct and incorrect classifications.”**